# Fault Diagnosis of Rotating Machinery under Noisy Environment Conditions Based on a 1-D Convolutional Autoencoder and 1-D Convolutional Neural Network

**DOI:** 10.3390/s19040972

**Published:** 2019-02-25

**Authors:** Xingchen Liu, Qicai Zhou, Jiong Zhao, Hehong Shen, Xiaolei Xiong

**Affiliations:** School of Mechanical Engineering, Tongji University, Shanghai 201804, China; qczhou@tongji.edu.cn (Q.Z.); jiong.zhao@tongji.edu.cn (J.Z.); 1510290@tongji.edu.cn (H.S.); xiong@tongji.edu.cn (X.X.)

**Keywords:** intelligent diagnosis, anti-noise diagnosis, convolutional neural network, convolutional autoencoder, rotating machinery

## Abstract

Deep learning methods have been widely used in the field of intelligent fault diagnosis due to their powerful feature learning and classification capabilities. However, it is easy to overfit depth models because of the large number of parameters brought by the multilayer-structure. As a result, the methods with excellent performance under experimental conditions may severely degrade under noisy environment conditions, which are ubiquitous in practical industrial applications. In this paper, a novel method combining a one-dimensional (1-D) denoising convolutional autoencoder (DCAE) and a 1-D convolutional neural network (CNN) is proposed to address this problem, whereby the former is used for noise reduction of raw vibration signals and the latter for fault diagnosis using the de-noised signals. The DCAE model is trained with noisy input for denoising learning. In the CNN model, a global average pooling layer, instead of fully-connected layers, is applied as a classifier to reduce the number of parameters and the risk of overfitting. In addition, randomly corrupted signals are adopted as training samples to improve the anti-noise diagnosis ability. The proposed method is validated by bearing and gearbox datasets mixed with Gaussian noise. The experimental result shows that the proposed DCAE model is effective in denoising and almost causes no loss of input information, while the using of global average pooling and input-corrupt training improves the anti-noise ability of the CNN model. As a result, the method combined the DCAE model and the CNN model can realize high-accuracy diagnosis even under noisy environment.

## 1. Introduction

Reliability is one of the most significant aspects in evaluating mechatronic systems and rotating machinery accounts for most of the reliability problems [1]. To eliminate safety risks, property and customer satisfaction loss to the maximum extent, the intelligent maintenance of rotating machinery is necessary. The superior maintenance method— Condition-Based Maintenance (CBM)—which takes maintenance actions based on condition monitoring, has become the preferred choice instead of the run-to-failure maintenance and the time-based preventive maintenance [2,3]. The core idea of CBM is to assess the degradation of mechatronic systems through diagnosis and prognosis methods using measured signals, and to take maintenance actions only when there is evidence of abnormal operating states [4]. As one of the key components of CBM, intelligent fault diagnosis has captured increasing attention. The typical intelligent diagnosis methods usually encapsulate two distinct blocks of feature extraction and classification, in which feature extraction is realized by signal processing techniques (fast Fourier transform, wavelet transform, etc.) and classification by machine learning methods (Artificial Neural Network, Support Vector Machine, etc.) [5,6]. However, the manual feature extraction customized for specific issues relies heavily on expert knowledge and may not be suitable for other cases, so lots of studies have turned to the end-to-end diagnosis, fusing the feature extraction and classification into a single learning body.

Because of their multilayer-structure, deep learning methods are powerful in both feature extraction and classification and are considered as the most effective end-to-end approaches. In recent years, deep learning has not only made great breakthroughs in the field of computer vision [7], natural language processing [8] and speech recognition [9], but some methods, such as deep brief network (DBN) [10], CNN [11] and stacked autoencoder (SAE) [12], have also found their way into CBM diagnosis applications. Among these methods, the CNNs hold the potential to be the end-to-end approaches because of the powerful feature learning ability of the convolutional structure. Since CNN was proposed by LeCun in 1998 and applied in handwriting recognition [13], lots of CNN models have been presented, such as AlexNet [14], VGGnet [15], GoogleNet [16] and ResNet [17]. With the 1D-to-2D conversion of vibration signals or 1-D convolutional structure, the CNN models have successfully applied in fault diagnosis directly using raw signals. Long et al. presented a 2-D convolutional neural network based on LeNet-5 through converting raw signals to 2-D images [18]. In 2016 [19], Turker et al. proposed a 1-D convolutional neural network with three improved convolutional layers and two fully-connected layers, which takes raw current signals as input and achieves 97.2% accuracy. In [20], a 2-D convolutional neural network with single convolutional layer and two fully-connected layers is proposed for fault diagnosis of gearbox.

Although lots of studies based on CNN have achieved high diagnosis accuracy, most of these prior studies focus on proposing new CNN models to speed up processing [19] and improve diagnostic accuracy [18,20], which demonstrate the effectiveness of the proposed models only using the noise-free experimental data. However, the signals measured in real industrial environments are always mixed with noise due to the complex operation conditions, and the CNN models with large numbers of parameters brought by the multilayer-structure are easy to overfit. As a result, depth models with excellent performance under experimental conditions may severely degrade in real industrial applications. In recent years, some training tricks (e.g., convolutional kernel dropout and ensemble learning [21]) and structure improvements (e.g., add AdaBN layers [22]) have been proposed to improve the anti-noise diagnosis ability of the deep CNN models, however, it is not enough to improve only the anti-noise ability of the model. On the one hand, the improvements proposed in the prior studies are used for reducing the overfitting of CNN models, but may not be that effective in anti-noise diagnosis. On the other hand, it is difficult to diagnose accurately only by improving the anti-noise ability when the signal is mixed with a lot of noise. Therefore, in order to solve the problem, not only the anti-noise ability should be improved, but also the noise mixed in raw signal should be removed as much as possible. In the area of image denoising, deep learning methods are widespread. In 2018 [23], a method based on deep CNN was proposed, which is trained by the image with non-fixed noise masks. Mao et al. proposed a deep DCAE-based method aiming at image denoising and super-resolution [24]. However, these methods are only applied in denosing of 2-D or 3-D images, and to the best of our knowledge, no research on similar methods aiming at the denoising of 1-D vibration signals has been reported so far.

In this paper, a novel method combining a 1-D denoising convolutional autoencoder (DCAE-1D) and a 1-D convolutional neural network with anti-noise improvement (AICNN-1D) is proposed to address the above problem. The former is used for noise reduction of raw vibration signals and the latter for fault diagnosis using the de-noised signals output by DCAE-1D. In summary, the contributions of this paper can be listed as follows:
(1)This paper proposes an effective denoising model based on 1-D convolutional autoencoder named DCAE-1D, which works pretty well directly on raw vibration signals with simple training.(2)This paper proposes an improved 1-D convolutional neural network named AICNN-1D for fault diagnosis, which applies global average pooling and is trained with randomly corrupted signals to improve the anti-noise ability of the model.(3)Both DCAE-1D and AICNN-1D model can work directly on vibration signals due to the use of 1-D convolution and 1-D pooling. Consequently, the method combining the two models can realize the end-to-end diagnosis.

The rest of this paper is organized as follows: a brief introduction of DCAE and CNN is provided in Section 2. In Section 3, the intelligent models based on 1-D convolutional structure are proposed: DCAE-1D for denoising and AICNN-1D for feature learning and classification. Bearing and gearbox experiments carried out to validate the proposed methods are detailed in Section 4. Finally, the conclusions are drawn in Section 5.

## 2. Denoising Convolutional Autoencoder and Convolutional Neural Network

The architecture of DCAE and CNN are briefly introduced in this section. Just as their names imply, the convolutional layer is the key component of both DCAE and CNN, and the introduction will start with the convolutional operation.

### 2.1. Convolutional Operation

Convolutional operation processes the input data using convolution kernel and output the processed feature, which is called feature map. A convolutional layer usually contains multiple kernels for extracting rich features. The universal convolutional operation is described as Equation (1):(1)$$y_{i}^{l+1}(j)=K_{i}^{l}*x^{l}(j)+b_{i}^{l},$$where $K_{i}^{l}$ and $b_{i}^{l}$ denote the weights and bias of the *i*-th kernel in the *l*-th layer, $x^{l}(j)$ presents the *j*-th local region of layer *l*. The * denotes the dot product of the kernel and the local regions, $y_{i}^{l+1}(j)$ is output value of convolutional operation.

Convolutional operation is usually followed by a nonlinear transformation through activation function. The ReLU, as a commonly used activation function, is expressed as Equation (2):(2)$$a_{i}^{l+1}(j)=f\left(y_{i}^{l+1}(j)\right)=\max\{0,y_{i}^{l+1}(j)\},$$where $a_{i}^{l+1}(j)$ presents the activation of $y_{i}^{l+1}(j)$.

### 2.2. Denoising Convolutional Autoencoder

An autoencoder is a model that tries to minimize reconstruct error between the input data and the output data. It contains two key parts, the encoder and the decoder. The encoder maps input data into low-dimensional space while the decoder reconstructs the compressed data into the same space as the input data. Training with some constraints, sparsity constraint for example, the extracted features of the hidden layer are useful for further processing.

#### 2.2.1. Traditional Denoising Autoencoder

As shown in Figure 1, the traditional autoencoder is a neural network with three fully-connected layers, the first two layers make up the coder and the last two layers make up the decoder. The number of input neural units is the same with the output. The denoising autoencoder (DAE) is an autoencoder that receives a corrupted (e.g., by some form of noise) data sample as input and is trained to predict the original, uncorrupted data sample as its output. As a result, the autoencoder undoes this corruption rather than simply copies their input, which can be regarded as denoising.

Giving the input training samples $X={\left\{x^{m}\right\}}_{m=1}^{M}$ ($x^{m}\in R^{n\times 1}$) and the corresponding corrupted samples $\tilde{X}={\left\{{\tilde{x}}^{m}\right\}}_{m=1}^{M}$, where M presents the number of samples, the encoder maps the corrupted input vector $\tilde{X}$ into low-dimensional representation $H={\left\{h^{m}\right\}}_{m=1}^{M}$($h^{m}\in R^{q\times 1}$) through encoder function, which is given by Equation (3): scale to text size and align(3)$$H=f(X)=s_{f}(WX+b),$$where W and b are the encoder weights matrix and biases, $s_{f}$ presents the activation function of the encoder. The decoder then reconstructs the hidden representation through the decoder function, which is expressed as Equation (4):(4)$$\hat{X}=g(H)=s_{g}(W^{\prime }H+b^{\prime }),$$where $\hat{X}={\left\{{\hat{x}}^{m}\right\}}_{m=1}^{M}$ (${\hat{x}}^{m}\in R^{n\times 1}$) presents the reconstructed output of the decoder, $W^{\prime }$ and $b^{\prime }$ present the decoder weights matrix and biases, $s_{g}$ is the activation function of the decoder.

The training target of the autoencoder is to minimize the reconstruct error between the output $\hat{X}$ and original input X through optimizing the network parameters $\theta =\left\{W,b,W^{\prime },b^{\prime }\right\}$. The reconstruct error is described by the loss function, and the mean square error (MSE) function is usually adopted: bold font is for vectors only(5)$$L(\theta )=\sum_{i=1}^{M}{\left\|X-\hat{X}\right\|}^{2}=\sum_{i=1}^{M}{\left\|X-g\left(f(\tilde{X})\right)\right\|}^{2},$$

#### 2.2.2. Denoising Convolutional Autoencoder

Compared with traditional DAE, DCAE has the same basic structure of encoder and decoder but replaces the fully-connected layers with convolutional layers. Because the CNN with deep structure is easy to train, DCAE, as a kind of CNN, can improve the reconstruct ability by using deep structure. As shown in Figure 2, the encoder and the decoder of DCAE are both with three 1-D convolutional layers.

In encoder part, the convolutional operation reduces the dimension of input vectors and therefore is not possible to employ the conventional convolutional operation to reconstruct a volume with the same dimension as its input. As a solution, transposed convolutional operation [25], which often applies input padding, is used in the decoder.

### 2.3. Convolutional Neural Network

CNN is a special structure of Artificial Neural Network (ANN) with several convolutional and pooling layers, and then followed by one or more fully-connected layers. The pooling layer is usually used for down sampling of input feature maps and the fully-connected layers for classification. The max-pooling and average-pooling are most commonly used in pooling layers, which performs local max and average operations over the input features respectively. The max-pooling and average-pooling are expressed as follows:(6)$$p_{i}^{l+1}=\underset{(j-1)W+1<t<jW}{\max}\left\{q_{i}^{l}(t)\right\},j=1,2,\ldots ,Q,$$(7)$$p_{i}^{l+1}=\underset{(j-1)W+1<t<jW}{\mathrm{avg}}\left\{q_{i}^{l}(t)\right\},j=1,2,\ldots ,Q,$$where $q_{i}^{l}(t)$ is the value of *t*-th neuron in the *i*-th feature map of the *l*-th layer, W is the width of pooling filter, *j* denotes the *j*-th moving step of the filter, and $p_{i}^{l+1}$ presents the corresponding value in layer *l* + 1 output by the pooling operation.

Because of the sparse connectivity and weights sharing, convolutional layer usually has much less parameters than fully-connected layer, which reduces the risk of overfitting. However, due to the influence of classic CNNs, the currently proposed CNN models applied in fault diagnosis all adopt 2~3 fully-connected layers as classifier [18,19,20,21,22]. Although the fully-connected layers can classify effectively, it brings a lot of parameters (e.g., account for more than 90% parameters in LeNet-5 and AlexNet), which leads to the risk of overfitting. In recent years, the global average pooling is used as classifier instead of the fully-connected layers in some deep CNNs, such as GoogleNet [16], MobileNet [26] and ShuffleNet [27]. It greatly reduces model parameters and is proved to be a good classifier. However, this improvement has not been studied and validated in the field of mechanical fault diagnosis.

## 3. Proposed Intelligent Diagnosis Method

The proposed intelligent method consists of two key stages as denoising and diagnosis, which are related to two convolutional models, namely DCAE-1D and AICNN-1D. DCAE-1D is used for noise reduction of raw vibration signals and AICNN-1D for fault diagnosis using the de-noised signals output by DCAE-1D.

### 3.1. DCAE-1D

Since vibration signals are one-dimensional time series, DCAE-1D employs a denoising autoencoder with 1-D convolutional layers to process the noisy input. As a result, the model can directly take raw signals as input without any extra preprocessing. DCAE-1D consists of three convolutional layers as coder and three transposed convolutional layers as decoder. The model architecture is shown in Figure 2 and the details are outlined in Table 1.

The training and testing of DCAE-1D utilize the noisy signals mixed by raw signal and Gaussian noise. Considering the continuous change of the noise in industrial environment, the noisy signals feed into the model should be with different SNR, which is defined as the ratio of signal power to the noise power as Equation (8):(8)$${\mathrm{SNR}}_{\mathrm{dB}}=10\log_{10}\left(\frac{P_{\mathrm{signal}}}{P_{\mathrm{noise}}}\right),$$

To compose the noisy signals with different SNR, the noise with different power are superposed to the same original signal respectively in whole area. The noise power is calculated according to the power of the original signal and the specified SNR value. Then each composited noisy signal is divided into multi-segments, which compose an original dataset with the specified SNR. Each original dataset is divided into two parts in a certain proportion, namely the training set and the testing set. Then all the training sets are gathered and shuffled to form the training dataset for training DACE-1D, while all the testing sets are respectively used for model testing.

### 3.2. AICNN-1D

AICNN-1D also employs 1-D convolution and pooling layers to adapt to the processing of vibration signals. Feature learning is realized by alternating convolutional and max-pooling layers, while classification by global average pooling layer. Compared with the existing CNN diagnosis models, improvements made in AICNN-1D are as follows:
(1)In order to reduce the number of parameters and the risk of overfitting, AICNN-1D adopts global average pooling as classifier instead of the fully-connected layers. The global average pooling layer does not bring any parameters and is faster in forward and backward propagation.(2)AICNN-1D is trained with randomly corrupted signals for improving the anti-noise diagnosis ability. The “corrupt” refers to the random selection of some data points of a sample in a certain proportion and the zeroization of the points, which is very similar to the “dropout”. The proportion is called “corruption rate”. Vincent et al. point out that the random-corruption can create noise for the input signals, which makes the trained model performs better even when the training set and testing set have different distributions [28].

AICNN-1D consists of three convolutional layers, two max-pooling layers and a global average pooling layer. The model architecture is shown in Figure 3 and the details are outlined in Table 2.

### 3.3. Construction of the Proposed Method

As shown in Figure 4, construction of the proposed method can be divided into three stages: (1) Training of DCAE-1D with datasets that mixed with Gaussian noise, (2) training of AICNN-1D using randomly corrupted datasets and (3) diagnosis test of the combination of the trained DCAE-1D and AICNN-1D using noisy datasets.

The samples used to train DCAE-1D vary in SNR from −2 dB to 12 dB. Training parameters and hyper-parameters are set as follows: the initial network weights are set by Xavier initialization [29], the biases initial values are set to 0. The learning rate uses exponential decay with 0.0035 base learning rate and 0.99 decay rate, the maximum training step *N* is set to 5000. The signals fed to the model in each step are randomly chosen from training dataset and the batch size is set to 100. Training is stopped after reaching the maximum step.

Training of AICNN-1D employs corrupted signals and the corruption rate is randomly set between 0 and 0.8. Training parameters and hyper-parameters are set as follows: the network weights and biases use the same initialization as DCAE-1D, the learning rate uses exponential decay with 0.005 base learning rate and 0.99 decay rate. The maximum training step and the batch size are set the same as DCAE-1D.

After training, diagnosis test is done by the combination of the trained DCAE-1D and AICNN-1D. DCAE-1D is used for denoising while AICNN-1D for diagnosis with the de-noised signals output by DCAE-1D. The reconstruct error is taken to evaluate the denoising effect of DCAE-1D and the diagnosis accuracy is considered as the test result of the method combination. The test is carried for several times using test sets with different SNR ranging from −2 dB to 12 dB and the SNR in the same test is kept constant. The hardware configuration of training and testing is Intel i7-5820k + NVidia 1080ti while the software environment is Windows10 + Python + TensorFlow.

## 4. Validation of the Proposed Method

The proposed method is validated by two experiments: bearing and gearbox diagnosis experiments. Experiments are conducted on the testbed shown in Figure 5, which consists of a motor, a gearbox, three bearings and a magnetic powder brake from left to right. The three bearings are both single row roller bearings with the same type (N205) and the gearbox is a two-stage spur gear reducer. Radial load is added on the second bearing by screw mechanism and the axial load by magnetic powder brake. The parameters of the bearings and the gearbox are detailed in Table 3.

### 4.1. Case 1: Bearing Diagnosis Experiment

The monitoring object of bearing diagnosis experiment is the third bearing. During the experiment, radial load is set to 0 N and axial load is set to 0 N·m, and motor speed is set to 1500 rpm.

A monoaxial piezoelectric acceleration sensor is used to acquire vibration signals, and the sensitivity of the sensor is 99.8 mv/g. Considering that the sensor should be close to the vibration source and in the direction of maximum vibration, it is installed on the bearing seat in the vertical direction as shown in Figure 5. Sampling frequency is set to 20,480 Hz through data acquisition unit (UT3408FRS-ICP). The bearing contains four health states: (1) normal condition (N), (2) inner race fault (IF), (3) outer race fault (OF) and (4) roller fault (RF). Wire cutting is used to process the normal bearing to simulate different faults. As shown in Figure 6, each fault has three different fault widths of 0.18 mm, 0.36 mm and 0.54 mm and has the same fault depth of 0.3 mm. As a result, the bearing contains 10 health conditions in total.

Corresponding to 10 bearing health conditions, each contains 3000 samples and the bearing dataset has 30,000 samples in total. The division of dataset is shown in Table 4: two-thirds of the dataset are used for training and the left for testing. Every data sample contains 2000 data points, so the input dimensions of DCAE-1D and AICNN-1D are both 2000 and the output dimensions are 2000 and 10, respectively. The samples used to train DCAE-1D are mixed with Gaussian noise and vary in SNR ranging from −2 dB to 12 dB, the samples used to train AICNN-1D are corrupted (dropout) with the corrupt rate randomly set between 0 and 0.8. After training, first test the denoising effect of the trained DCAE-1D using several test sets with different SNR from −2 dB to 12 dB, and then test the diagnosis effect of AICNN-1D using the corresponding de-noised test sets output by DCAE-1D.

#### 4.1.1. Validation of the Denoising and Diagnostic Effects of the Proposed Method

The experimental results can be divided into two parts: the reconstruct error of DCAE-1D and the diagnosis accuracy of AICNN-1D, in which the former is used to validate the denoising effect of DCAE-1D and the latter to validate the diagnostic effect of the method combination under noisy environment.

The traditional DAE is used to compare with the proposed DACE-1D in denoising effect under conditions of different SNR. The input, hidden and output dimensions of the DAE are set to 2000, 1000 and 2000, respectively, and ReLU is used as the activation function. As shown in Figure 7 and Table 5, the reconstruction error of DCAE-1D is much less than that of DAE under all SNR conditions. In particular, the reconstruct error of DCAE-1D is less than 0.025 but that of DAE is greater than 0.055 when SNR = −2 dB. The result proves that (delete “the proposed”) DCAE-1D is much more effective in noise reduction than DAE, especially under low SNR conditions. The reasons are analyzed as follows. Firstly, DCAE-1D is superior to DAE in reconstruction ability owning to the deeper structure. Secondly, DCAE-1D has much less parameters (2 × (15 × 16 + 10 × 32 × 16 + 5 × 32 × 64) = 31,200) than that of DAE (2000 × 1000 × 2 = 4,000,000), which makes it not easy to over fit under noisy conditions. In addition, under the SNR greater than 7 dB, the reconstruct errors of DCAE-1D are close to 0.001 while that of DAE are greater than 0.012, which indicates that the former almost causes no loss of input information in comparison with the latter.

Figure 8 shows the original signal, the noisy signal, the de-noised signal of DCAE-1D and the de-noised signal of DAE under the condition of SNR = −1 dB and roller fault (0.54 mm). It can be seen that the fault characteristics in original signal are masked by the added Gaussian noise. After the denoising by DCAE-1D or DAE, the noise mixed in the original signal is obviously reduced. However, the noise cannot be eliminated completely, and the result shows that the de-noised signal of DCAE-1D leaves much less noise than that of DAE.

The diagnostic results of the method combination of AICNN-1D and DCAE-1D under conditions of different SNR are shown in Figure 9 and Table 6. It can be seen that the accuracy is up to 96.65% even when SNR= −2 dB and greater than 99% when SNR > −1 dB, which proves that the proposed method combination can achieve high-accuracy diagnosis even under low SNR conditions.

Besides, the “AICNN-1D + DAE” combination and the AICNN-1D without denoising are compared with the proposed “AICNN-1D + DCAE-1D” combination as follows. The diagnosis accuracy of the “AICNN-1D + DAE” combination and AICNN-1D without denoising drops to 94.23% and 91.64% respectively when SNR = −2 dB. The result shows that the denoising step is significant to the accuracy improvement under low SNR conditions and DCAE is much more effective in noise reduction than DAE. In addition, other two points of the results are worth to be noticed: Firstly, under the conditions of SNR > 2 dB, the diagnosis accuracy of the “AICNN-1D + DAE” combination is even lower than AICNN-1D without denoising. The reason for this phenomenon is that DAE causes information loss, which outweighs the gain of the denoising. Secondly, when SNR = −2 dB, even though the diagnosis accuracy of AICNN-1D without denoising is less than the other two method combinations, but considering that there is no denoising, the accuracy of 91.64% is pretty well. It indicates that AICNN-1D has the capability of anti-noise diagnosis to some extent.

Figure 10 presents the confusion matrix of the diagnosis result under the condition of SNR = −2 dB and the overall accuracy is 96.65%. It shows that the accuracy of normal condition is the lowest one, which is 94.5%. Besides, the IF-0.018, OF-0.018 and RF-0.018, regarded as slight fault conditions, are 94.8%, 95.4% and 95.7% in accuracy respectively, which are lower than the rest health conditions. The results demonstrate that the normal and slight fault conditions are more likely to be misclassified under low SNR conditions. Furthermore, it can be found from Figure 10 that the normal condition is likely to be misclassified into the slight fault conditions and vice versa. As an instance, the misclassification rates of normal to IF-0.018, normal to OF-0.018 and normal to RF-0.018 are 1.7%, 1.8% and 1.3% respectively and that of normal to other conditions are lower obviously.

#### 4.1.2. Validation of Input-Corrupt Training and Global Average Pooling

In order to test the effects of the input-corrupt training and the global average pooling applied in AICNN-1D, the model with “input-corrupt training + global average pooling” (model A), “clean-input training + global average pooling” (model B) and “input-corrupt + fully-connected layers” (model C) are compared in Figure 11 and Table 7. When SNR < 8 dB, the proposed model A is superior to model B and C in accuracy. Especially, compared with model A, the accuracy of model B and C drops to 93.54% and 94.26% from 96.65% under the condition of SNR= −2 dB. The results prove that the input-corrupt training and the global average pooling are both effective in improving the anti-noise ability of the model. On the one hand, the input-corrupt training is a regularization method, on the other hand, the application of global average pooling dramatically reduces the number of model parameters. Both of them prevent the model from overfitting, which improves the performance in the diagnosis of noisy signals. However, the accuracies of the three models are very close under high SNR conditions (SNR > 9 dB). For example, the accuracies of the three models are 99.78%, 99.75% and 99.78% when SNR = 12 dB. The reason is that all the three models are not easy to over fit in processing the signals with little noise.

#### 4.1.3. Optimization of the DACE-1D and AICNN-1D

When fine turning the proposed models, some configurations and parameters have significant influence on their performance. Among these factors, two of them influence the performance obviously, namely the depth of DCAE-1D and the kernel width of the first convolution layer of AICNN-1D. As outlined in Table 8, the reconstruction errors of DCAE-1D show a trend of ascending after descending with the increase of the depth. For example, when SNR = −2 dB, the reconstruction errors with the depth of 2, 4, 6 and 8 are 0.0364, 0.0256, 0.0243 and 0.0249, respectively. The result demonstrates that the selection of the model depth is significant to the denoising effect, and in this case, the optimum depth is 6.

The diagnosis accuracies of AICNN-1D with different kernel width of the first convolutional layer are outlined in Table 9. The result shows that the diagnosis accuracy rises with the increase of kernel width before reaching 15, but when greater than 15, the increase of kernel width almost has no effect on the model performance. Considering the increase of the kernel width brings more parameters and higher computing cost, the width of 15 is the optimal choice.

#### 4.1.4. Comparisons with the Existing Models

Two baseline models are compared with the proposed method. The first one, proposed by Chen et al. in 2017 [30], is based on the stacked denoising autoencoder (SDAE). The SDAE-based model establishes the deep hierarchical structure with unsupervised layer-by-layer pre-training and supervised fine turning. The dropout and sparsity representation are used to improve the anti-noised ability of the model. In the original paper, the input neurons of each layer are 200, 100, 50, 25 and 10. However, in order to compare with the method proposed in this paper, the input neurons of each layer are set to 2000, 1000, 500, 100 and 10, and other parameters and configurations are kept unchanged. The second one is named WDCNN proposed by Zhang et al. in 2017 [22]. WDCNN adopts five 1-D convolutional layers, five max-pooling layers and two fully-connected layers. In order to improve the anti-noise ability, AdaBN layers and wide first-layer kernels are employed. The experimental results are shown in Figure 12.

Without denoising, both the SDAE-based model and WDCNN are inferior to the proposed method under the conditions of low SNR. For instance, when SNR< 3 dB, the accuracies of the two models are less than 95%, but that of the proposed method is greater than 96.65%, especially, when SNR= −2 dB, the accuracies of the two models are 72.14% and 82.13% respectively. With the denoising by DCAE-1D, the accuracies of the SDAE-based model and WDCNN are both obviously improved, especially under the conditions of low SNR. As an example, the accuracy of the SDAE-based model rises from 72.34% to 89.12%, and that of WDCNN rises from 82.13% to 92.24% when SNR= −2dB. The results imply that in order to diagnosis accurately under low SNR conditions, the denoising is indispensable. In addition, with the same denoising preprocessing, the proposed AICNN-1D still performs better than the SDAE-based model and WDCNN obviously when SNR< 3 dB. It proves that AICNN-1D is superior to the other two models in anti-noise ability, for the parameters of AICNN-1D are much less than the other two models and the using of input-corrupt training.

The classic shallow models of the BP neural network [31] and the SVM [32] are also adopted to compare with the proposed method. The input, hidden and output dimensions of the BP neural network is 2000, 1000 and 10, and the sigmoid function is used as activation function. The SVM model uses Gaussian kernel function. Various configurations for the BP neural network and the SVM are explored the configurations that achieved the best performance are empirically selected. As shown in Figure 12, even with the denoising by DCAE-1D, these two models both perform much worse than the three models above under all SNR conditions. For example, the accuracies of the BP neural network and the SVM are only 87.68% and 92% respectively even when SNR = 12 dB. The results indicate that the shallow models cannot extract discriminative features from vibration signals and cannot be regarded as effective end-to-end methods.

The comparison results of computational time between the existing models and the proposed models are shown in Table 10. It can be seen that the training of DCAE-1D and AICNN-1D costs 106.88 ± 3.75 s and 53.25 ± 2.79 s respectively, and the testing costs 0.79 ± 0.05 s and 0.42 ± 0.06 s. Compared with the existing deep method of the SDAE-based model and WDCNN, which costs 98.64 ± 2.58 s and 0.71 ± 0.12 s in training respectively and costs 0.71 ± 0.12 s and 0.84 ± 0.08 s in testing, the proposed method combination consumes slightly more time in training and testing. In addition, the shallow models of BP neural network and SVM are much less time-consuming in both training and testing.

### 4.2. Case 2: Gearbox Diagnosis Experiment

The monitoring object of gearbox diagnosis experiment is the driven gear and the experiment conditions are the same as the bearing diagnosis. During experiment, all the bearings are kept in Normal states. The acceleration sensor and its sampling frequency are the same as the bearing experiment configuration. Considering the up cover of the gearbox is smooth in surface and receives the vibration of the gear directly, the sensor is installed on the center of the up cover of gearbox in the vertical direction, as shown in Figure 5. The driven gear contains five health states as shown in Figure 13: (1) normal condition (N), (2) tooth pitting fault (TPF), (3) tooth break fault (TBF), (4) tooth crack fault (TCF) and (5) tooth wear fault (TWF). Five normal gears are used in the experiment named as gears A~E, respectively. Electrical discharge machining is employed to process a single tooth on gear B to simulate tooth pitting fault, while wire cutting is utilized to process a single tooth on gear C and D to simulate tooth crack and tooth break fault. Besides, all the teeth on gear E are ground by gear grinding machine to simulate tooth wear fault. Configurations of the models and division of datasets are the same as bearing experiment except setting output dimension of AICNN-1D to 5.

#### 4.2.1. Validation of Denoising and Diagnostic Effects under Noisy Environment

The denoising test result of DCAE-1D and DAE under different SNR is shown in Figure 14 and Table 11. Consistent with the bearing experiment result, the reconstruct error of DCAE-1D is also much less than that of DAE under the same SNR. In particular, the reconstruct error is close to 0 when SNR is greater than 8 dB. The experimental result demonstrates that DCAE-1D is effective in denoising of gearbox vibration signals and it almost cause no loss of input information.

The diagnostic results of “AICNN-1D + DCAE-1D”, “AICNN-1D + DAE” and AICNN-1D without denoising are shown in Figure 15 and Table 12. As illustrated by the figure and the table, the “AICNN-1D + DCAE-1D” is superior to the other two in diagnosis accuracy under all SNR conditions, which is consistent with the bearing experiment. Especially, the accuracy is up to 97.25% even when SNR = −2 dB and greater than 99% when SNR > −1 dB. It proves that the proposed method combination can achieve high-accuracy diagnosis of gear even under low SNR conditions. In addition, the accuracy of “AICNN-1D + DAE” is even lower than AICNN-1D when SNR is higher than 6 dB, which demonstrates that the information loss caused by DAE outweighs the gain of the denoising.

#### 4.2.2. Validation of Partial Varying SNR Situation

In the previous experiments, the denoising of DCAE-1D and the diagnosis of AICNN-1D are tested by several datasets with different SNR. Although the SNR varies between the different datasets, it remains constant in the same dataset. Furthermore, in order to simulate the real noise that continuous changing in industrial environment as exactly as possible, the dataset with partially varying SNR is used to test the denoising of DCAE-1D and the diagnosis of AICNN-1D. To be specific, the dataset is obtained through the following steps: Firstly, divide each sample of the noise-free test set into 10 segments in sequence, and then assign each segment a random SNR value between −2 dB to 12 dB. Secondly, the Gaussian noise of specific power is superposed to each segment according to the power and the assigned SNR value of the signal segment. Finally, recombine every 10 noisy segments into one sample in the original order.

The dataset is generated 20 times through the steps above. Correspondingly, each generated dataset is used to test the denoising of DCAE-1D and the diagnosis of AICNN-1D independently. The statistical results of the 20 test are as follows: the reconstruction error of DCAE-1D is 0.0063 ± 0.0005 and the accuracy of AICNN-1D is 99.34% ± 0.17%. The results demonstrate that the method combination of DACE-1D and AICNN-1D is also effective under the situation of partial varying SNR.

#### 4.2.3. Feature Learning Visualization

In order to validate the adaptive feature learning ability of the proposed AICNN-1D under noisy environment conditions, the t-distributed stochastic neighbor embedding (t-SNE) is used to visualize the learned features of the input layer and the three convolution layers under the condition of SNR = 0 dB. As the experimental results shown in Figure 16, the features of the input layer and the first convolutional layer are of poor discrimination. Nonetheless, as the depth increases, the features learnt by the convolutional layers become more and more discriminative. The learned features of the last convolutional layer are almost with no overlap between different fault types, which proves the model can adaptively learn effective features for accurate fault diagnosis even under the condition of SNR = 0 dB. It is noteworthy that, the learned features of a few TCF and TWF samples in the last convolutional layer overlap with the set of normal features, which indicates that the TCF and TWF samples may be misclassified into the label of normal under the low SNR conditions.

## 5. Conclusions

This paper proposes a combined method of 1-D denoising convolutional autoencoder named DCAE-1D and 1-D convolutional neural network named AICNN-1D, which aims at addressing the fault diagnosis problem under noisy environment. Compared with the existing method that only improves the anti-noise ability [21,22,30], the extra noise reduction by DACE-1D is introduced in the combined method. With the denoising of raw signals, AICNN-1D with anti-noise improvements is then used for diagnosis. The method combination is validated by the bearing and gearbox datasets mixed with Gaussian noise. Through the analysis of the experimental results in Section 4, conclusions are drawn as follows:

The proposed DCAE-1D with deep structure of three convolutional and three transposed convolutional layers is much more effective in denoising than the traditional DAE. The reconstruct errors of DCAE-1D in bearing and gearbox datasets are only 0.24 and 0.0109 even when the SNR= −2 dB, while that of DAE are 0.0559 and 0.01824 respectively. Besides, under the high SNR (> 4dB) conditions, the reconstruct errors are close to 0, which indicates that DACE-1D almost causes no loss of input information.

With the denoising of DCAE-1D, the diagnosis accuracies of AICNN-1D are 96.65% and 97.25% respectively in bearing and gearbox experiments even when SNR = −2 dB. When SNR> −1 dB, the diagnosis accuracies are greater than 99% in the two experiments. The result indicates that the proposed method combination realizes high diagnosis accuracy under low SNR conditions. In the bearing experiment, the AICNN-1D with “clean-input training + global average pooling” and “input-corrupt + fully-connected layers” drops to 93.54% and 94.26% in accuracy when SNR = −2 dB, compared with the proposed AICNN-1D. The results demonstrate that the using of input-corrupt training and global average pooling is effective in improving the anti-noise ability. Compared with the existing methods of the SDAE-based model and WDCNN presented in 2017, the proposed method shows its superiority under low SNR conditions. For instance, the diagnosis accuracies of the SDAE-based model and WDCNN are 72.34% and 82.13% when SNR = −2 dB in the bearing experiment. With the denoising by DCAE-1D, the diagnosis accuracies of the two models rise to 89.12% and 92.24% respectively, proving that both the denoisng of input signals and the anti-noise ability of the model are indispensable in accurate diagnosis under low SNR conditions.

In this paper, the load and rotating speed are kept constant during experiment, which is a simplification of the actual conditions in industrial applications. It will be more valuable to extend the proposed method to the cases with more complex conditions (e.g., variable speed and load) in the future. Besides, this paper focus on the diagnosis problem of industrial applications and validate the method with massive experimental data. However, the faulty sets in industrial systems are generally not readily to acquire. As the possible solutions, two kinds of methods are worth noticing: data augmentation and data fusion. The former focus on generating more samples on the basis of existing samples, such as data overlapping [21] and Generative Adversarial Network (GAN) methods [33]. The latter studies fusing the data of multiple kinds of sensors, which can be grouped into three main approaches: data-level fusion [34], feature-level fusion [35] and decision-level fusion [36].

## Figures and Tables

**Figure 1 sensors-19-00972-f001:**
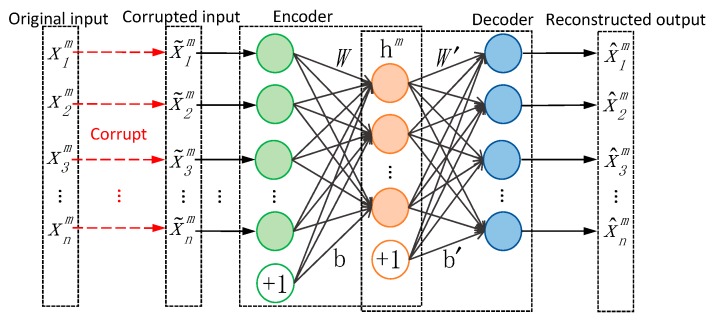
Structure of traditional denoising autoencoder: the learning target is minimizing the error between the reconstructed output and the original input.

**Figure 2 sensors-19-00972-f002:**
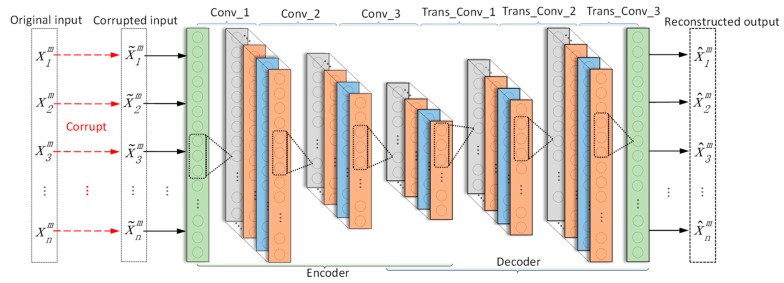
Structure of a 1-D denoising convolutional autoencoder.

**Figure 3 sensors-19-00972-f003:**
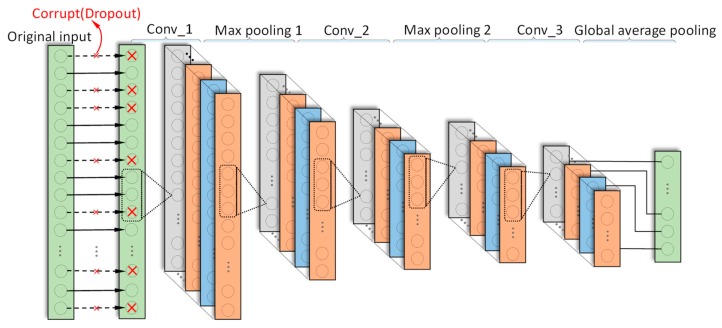
Architecture of AICNN-1D.

**Figure 4 sensors-19-00972-f004:**
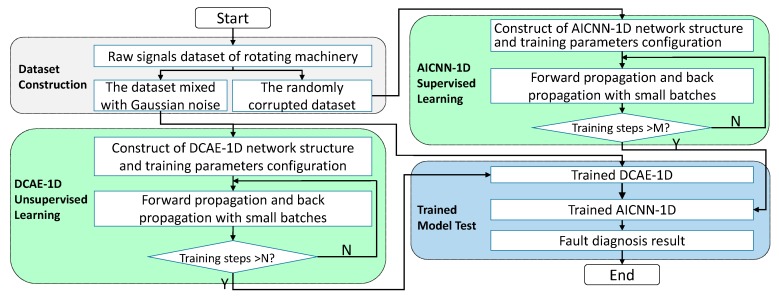
Flowchart of the proposed method.

**Figure 5 sensors-19-00972-f005:**
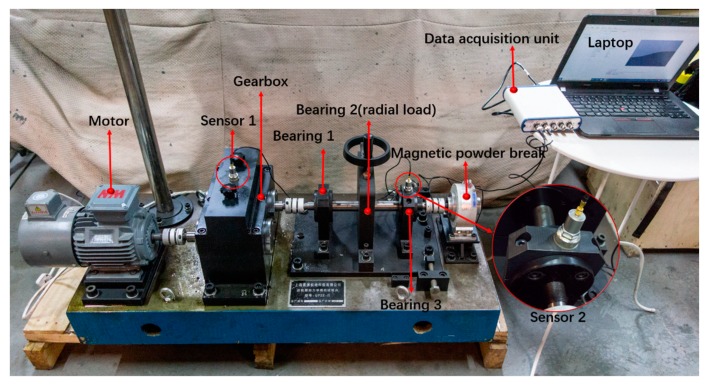
Fault diagnosis testbed of rotating machinery.

**Figure 6 sensors-19-00972-f006:**
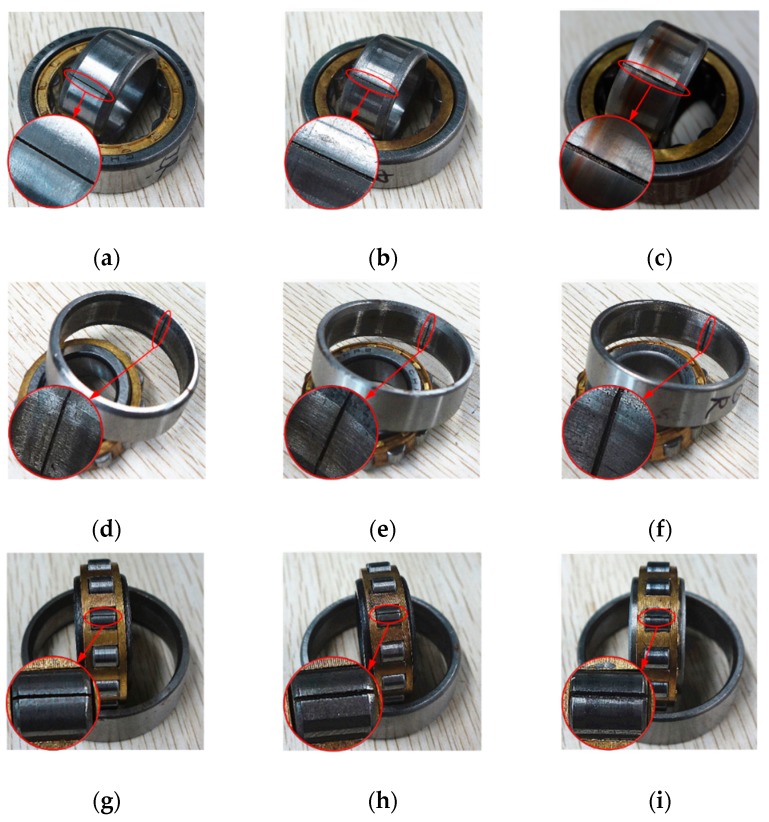
Bearing with different fault types and fault sizes: (**a**) IF, 0.18 mm, (**b**) IF, 0.36 mm, (**c**) IF, 0.54 mm, (**d**) OF, 0.18 mm, (**e**) OF, 0.36 mm, (**f**) OF, 0.54 mm, (**g**) RF, 0.18 mm, (**h**) RF, 0.36 mm, (**i**) RF, 0.54 mm.

**Figure 7 sensors-19-00972-f007:**
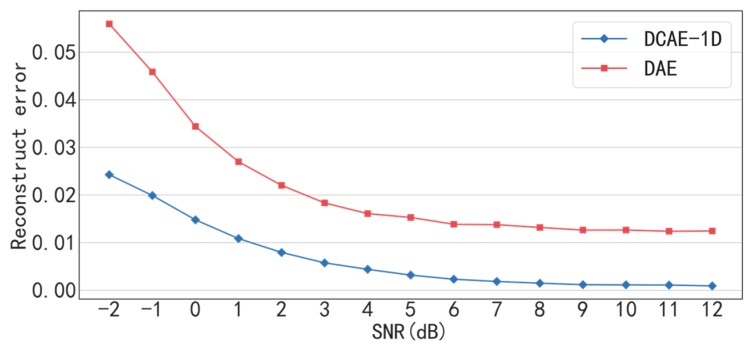
Reconstruction errors of DCAE-1D and DAE under different SNR.

**Figure 8 sensors-19-00972-f008:**
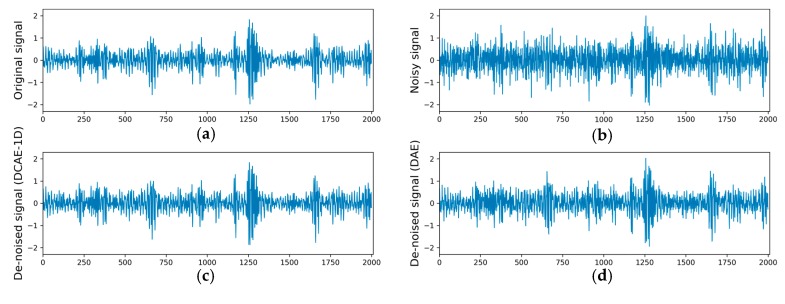
Figures of signals: (**a**) original signal, (**b**) original signal mixed with Gaussian noise, (**c**) de-noised signal of DACE-1D and (**d**) de-noised signal of DAE under the condition of SNR = −1 dB and roller fault (0.54 mm).

**Figure 9 sensors-19-00972-f009:**
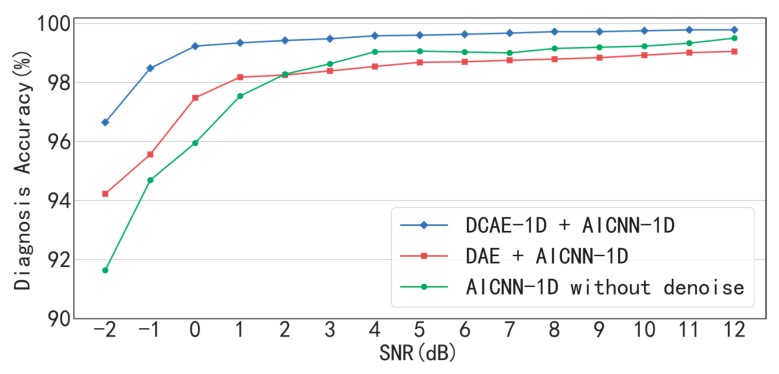
Diagnosis accuracy of AICNN-1D with different denoising processes.

**Figure 10 sensors-19-00972-f010:**
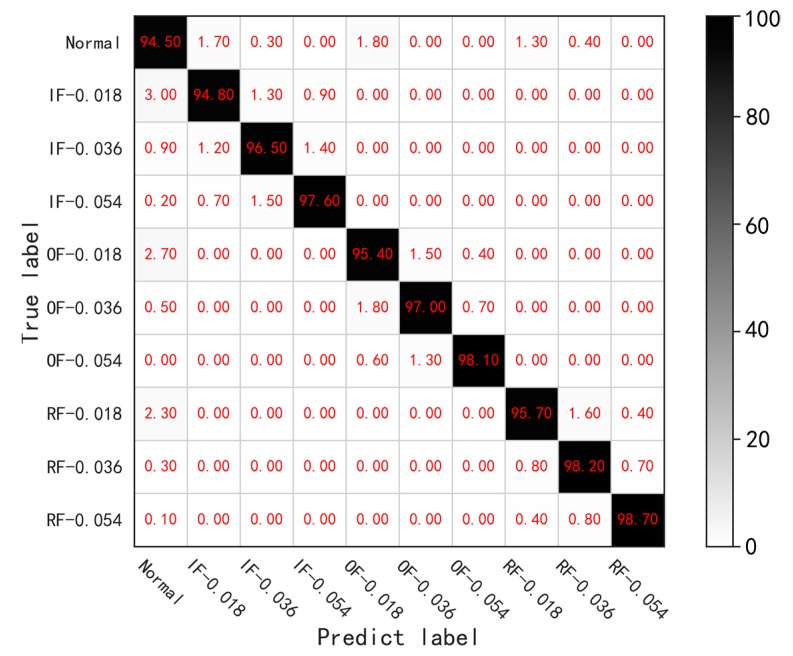
The confusion matrix of the diagnosis result in the condition of SNR= −2 dB.

**Figure 11 sensors-19-00972-f011:**
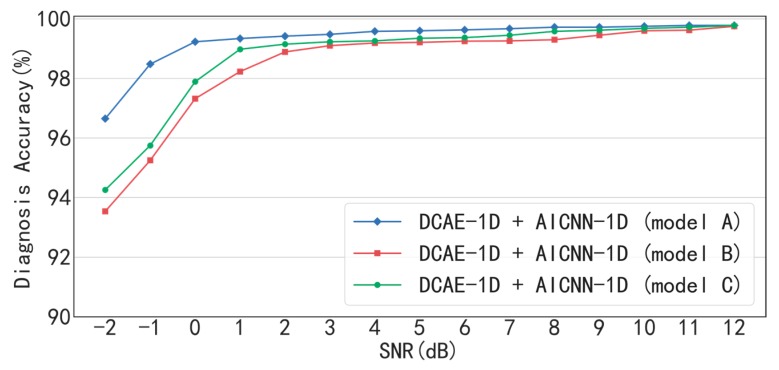
Diagnosis accuracy of AICNN-1D with input-corrupt training and global average pooling, with clean-input training and with fully-connected layers.

**Figure 12 sensors-19-00972-f012:**
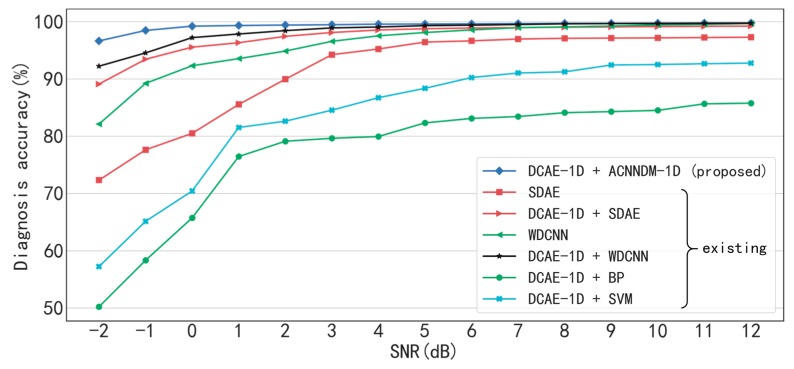
The comparison between the proposed method and existing models.

**Figure 13 sensors-19-00972-f013:**
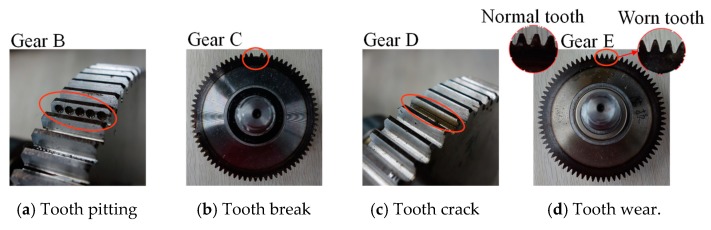
The gears with different fault type.

**Figure 14 sensors-19-00972-f014:**
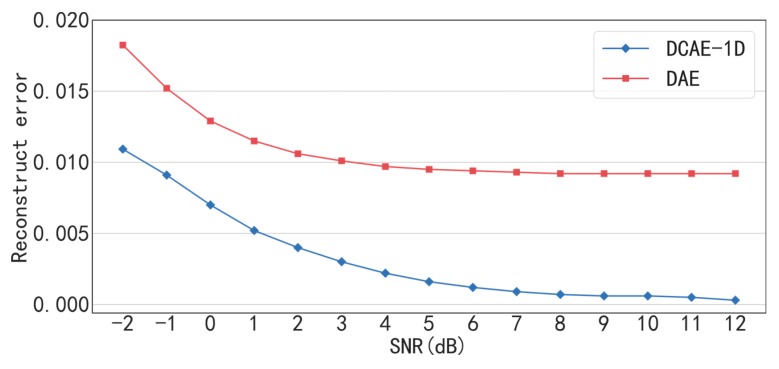
Reconstruction error of DCAE-1D and DAE under different SNR.

**Figure 15 sensors-19-00972-f015:**
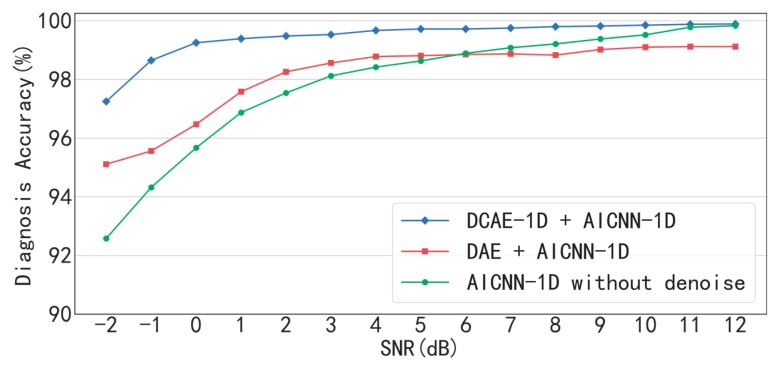
Diagnosis accuracy of AICNN-1D with different denoising process (delete something).

**Figure 16 sensors-19-00972-f016:**
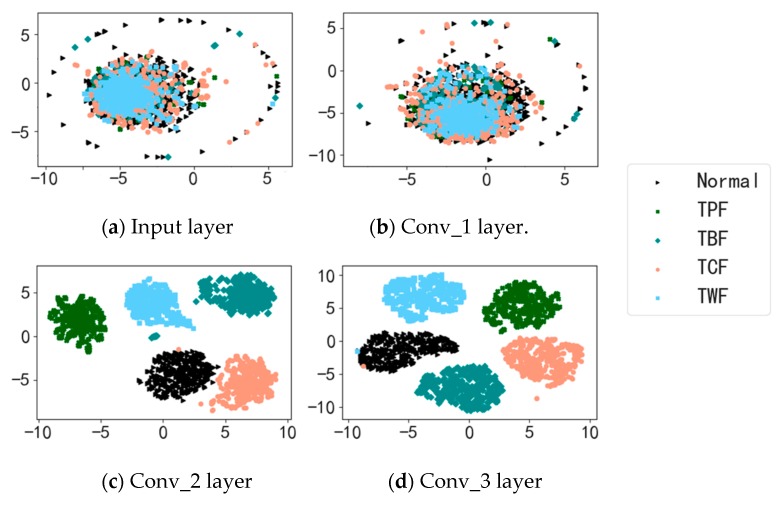
Feature learning visualization with t-SNE.

**Table 1 sensors-19-00972-t001:** Details of DCAE-1D.

Layer	Kernel Size	Kernel Number	Stride	Activation Function	Output Size	Padding
Conv_1	15	16	2	LeakyReLU	1000 × 16	Yes
Conv_2	10	32	2	LeakyReLU	500 × 32	Yes
Conv_3	5	64	2	LeakyReLU	250 × 64	Yes
Trans_conv_1	5	32	2	LeakyReLU	500 × 32	Yes
Trans_conv_2	10	16	2	LeakyReLU	1000 × 16	Yes
Trans_conv_3	15	1	2	None	2000	Yes

**Table 2 sensors-19-00972-t002:** Details of AICNN-1D.

Layer	Kernel Size	Kernel Number	Stride	Activation Function	Output Size	Padding
Conv_1	15	16	7	ReLU	286 × 16	Yes
Max-pooling 1	2	16	2	None	143 × 16	Yes
Conv_2	10	32	5	ReLU	29 × 32	Yes
Max-pooling 2	2	32	2	None	15 × 32	Yes
Conv_3	5	H^1^	2	ReLU	8 × H	Yes
Global average pooling	8	H	1	None	H	No

^1^ H represents the number of classification labels.

**Table 3 sensors-19-00972-t003:** Parameters of bearing and gearbox.

Parameters		Value
**Bearing:**	Pitch diameter	38.5 (mm)
	Roller diameter	6.5 (mm)
	Roller number	13
**Gearbo** **x:**	Gear module	2 (mm)
	Face width	20 (mm)
	Pressure angle	20°
	Driving/driven gear teeth number	55/75
	Gear clearance	0.5 (mm)

**Table 4 sensors-19-00972-t004:** Description of bearing dataset.

Health Conditions	Train Samples	Test Samples	Classification Label
N	2000	1000	0
IF (0.18/0.36/0.54)	2000/2000/2000	1000/1000/1000	1/2/3
OF (0.18/0.36/0.54)	2000/2000/2000	1000/1000/1000	4/5/6
RF (0.18/0.36/0.54)	2000/2000/2000	1000/1000/1000	7/8/9

**Table 5 sensors-19-00972-t005:** Reconstruction errors of DCAE-1D and DAE under different SNR.

Method Combination	SNR(dB)							
	−2	0	2	4	6	8	10	12
DCAE-1D	0.0243	0.0148	0.0080	0.0044	0.0023	0.0015	0.0011	0.0009
DAE	0.0559	0.0344	0.0220	0.0161	0.0139	0.0132	0.0127	0.0125

**Table 6 sensors-19-00972-t006:** Diagnosis accuracy of AICNN-1D with different denoising processes.

Method Combination	SNR (dB)							
	−2	0	2	4	6	8	10	12
DCAE-1D + AICNN-1D	**96.65**	**99.23**	**99.42**	**99.58**	**99.63**	**99.72**	**99.75**	**99.78**
DAE + AICNN-1D	94.23	97.48	98.25	98.54	98.7	98.79	98.92	99.05
AICNN-1D without denoising	91.64	95.95	98.28	99.04	99.03	99.15	99.23	99.5

**Table 7 sensors-19-00972-t007:** Diagnosis accuracy of AICNN-1D with input-corrupt training and global average pooling, with clean-input training and with fully-connected layers.

Method	SNR (dB)							
	−2	0	2	4	6	8	10	12
DCAE-1D + AICNN-1D (model A)	**96.65**	**99.23**	**99.42**	**99.58**	**99.63**	**99.72**	**99.75**	**99.78**
DCAE-1D + AICNN-1D (model B)	93.54	97.32	98.89	99.19	99.25	99.30	99.60	99.75
DCAE-1D + AICNN-1D (model C)	94.26	97.89	99.15	99.26	99.37	99.58	99.68	99.78

**Table 8 sensors-19-00972-t008:** Reconstruction errors of DCAE-1D with different depth.

Depth of DCAE-1D	SNR (dB)							
	−2	0	2	4	6	8	10	12
2: 1 Conv + 1 Trans_conv	0.0364	0.0198	0.0095	0.0054	0.0038	0.0032	0.0029	0.0028
4: 2 Conv + 2 Trans_conv	0.0256	0.0153	**0.0080**	**0.0044**	0.0025	0.0017	0.0014	0.0011
6: 3 Conv + 3 Trans_conv	**0.0243**	**0.0148**	**0.0080**	**0.0044**	**0.0023**	**0.0015**	**0.0011**	**0.0009**
8: 4 Conv + 4 Trans_conv	0.0249	0.0151	0.0083	0.0047	0.0028	0.0020	0.0017	0.0016

**Table 9 sensors-19-00972-t009:** Diagnosis accuracy (%) of AICNN-1D with different kernel width of the first convolution layer.

Kernel Width	SNR (dB)							
	−2	0	2	4	6	8	10	12
5	96.48	99.16	99.28	99.35	99.50	99.62	99.60	99.60
10	96.50	99.22	99.36	99.52	99.58	99.68	99.72	99.74
15	96.65	**99.23**	**99.42**	99.58	**99.63**	**99.72**	99.75	**99.78**
20	**96.66**	99.22	99.40	99.60	**99.63**	99.70	99.76	**99.78**
25	96.65	**99.23**	99.41	**99.62**	99.62	99.70	**99.78**	**99.78**

**Table 10 sensors-19-00972-t010:** Computational time of the proposed models and the existing models.

Method	Training Time (5000 × 100 Samples)	Testing time (10000 Samples)
DCAE-1D + AICNN-1D	(106.88 ± 3.75) s + (53.25 ± 2.79) s	(0.79 ± 0.05) s + (0.42 ± 0.06) s
SDAE	98.64 ± 2.58 s	0.71 ± 0.12 s
WDCNN	126.27 ± 5.24 s	0.84 ± 0.08 s
BP	40.43 ± 0.93 s	0.36 ± 0.06 s
SVM	63.56 ± 0.78 s	0.57 ± 0.11 s

**Table 11 sensors-19-00972-t011:** Reconstruction error of DCAE-1D and DAE under different SNR.

Method	SNR (dB)							
	−2	0	2	4	6	8	10	12
DCAE-1D	**0.0109**	**0.0070**	**0.0040**	**0.0022**	**0.0012**	**0.0007**	**0.0006**	**0.0003**
DAE	0.01824	0.0129	0.0106	0.0097	0.0094	0.0092	0.0092	0.0092

**Table 12 sensors-19-00972-t012:** Diagnosis accuracy of AICNN-1D with different denoising process (delete something).

Method Combination	SNR (dB)							
	−2	0	2	4	6	8	10	12
DCAE-1D + AICNN-1D	**97.25**	**99.25**	**99.48**	**99.67**	**99.72**	**99.8**	**99.85**	**99.89**
DAE + AICNN-1D	95.11	96.47	98.26	98.78	98.85	98.83	99.1	99.12
AICNN-1D without denoising	92.58	95.67	97.54	98.42	98.89	99.21	99.52	99.83

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
