# Peer review of "Fault Diagnosis of Rotating Machinery under Noisy Environment Conditions Based on a 1-D Convolutional Autoencoder and 1-D Convolutional Neural Network"

_sensors, 2019, doi:10.3390/s19040972_

Reviewer 1 Report

Review report on manuscript number ID: sensors-430223 submitted to "sensors".

 Title: Fault diagnosis of rotating machinery under noisy environment based on 1-D convolutional autoencoder and 1-D convolutional neural network

 Authors: Xingchen Liu, Qicai Zhou, Jiong Zhao, Hehong Shen and Xiaolei Xiong

 This paper proposes a combined method of 1-D denoising convolutional autoencoder named DCAE-1D and 1-D convolutional neural network named AICNN-1D, addressing the problem of fault diagnosis under noisy environment. Both of the DCAE-1D and AICNN-1D can work directly on vibration signals because the using of 1-D convolution and 1-D pooling, so the combined method can realize the end-to-end diagnosis. The subject is interesting and relevant to the field of this journal.  

The language should be further improved by a native English speaker.

Τhe abstract does not provide the reader with information about the results obtained. A critical analysis of the results and explanation of trends is needed. It needs to be improved, as it is very cryptic in the current form, without numerical values for the results.

The authors should change all the sentences in the text which are written in the first plural (line 72, 76, 79, 84, 86, 93, 96, 99, 107, 176, 225, 260, 307, 321, 339, 344, 417, 432 and 434.  

In lines 244-245, “The vibration signals are acquired by acceleration sensor installed on bearing seat in vertical direction…..” The authors don’t give any information about the accelerometers. Are they triaxial or monoaxial? How the specific position was selected? Why vertical? What is the specific position? Give a photo.

In lines 248-249 the authors mention “As shown in Figure 6, each fault has three different fault width of 0.18mm, 0.36mm and 0.54mm respectively….”. Suggested to the authors to give some information about the faults. How that damages were caused. Maybe a photo. The same in Case study 2.

In line lines 243-244 the authors mention that “The radial load is set to 0 N and axial load is set to 0 N·m, while the motor speed is set to 1500 rpm”. All the measurements in the experimental system were carried out with any load? Why?

Conclusion: The Conclusions are very poor and are not suitable, should give more useful conclusions. Should include more numerical values for the results. The authors should rewrite more analytically this section.

Author Response

Dear Reviewer:

Thank you for your comments concerning our manuscript entitled “Fault diagnosis of rotating machinery under noisy environment based on 1-D convolutional autoencoder and 1-D convolutional neural network” (ID: sensors-430233). Those comments are all valuable and very helpful for revising and improving our paper, as well as the important guiding significance to our researches. We have studied comments carefully and have made correction which we hope meet with approval. Revised portion are marked in red in the paper. The main corrections in the paper and the responds to your comments are as flowing:

 Point 1: The language should be further improved by a native English speaker.

 Response 1: The language has been thoroughly checked and the improvements are listed as follows:

(1)Add definite articles in places, or change some indefinite articles to definite articles.

(2)Correct the misusing of punctuation, such as changing the comma (,) to (.), or changing the comma (;) to (.) or (,), et al.

(3)Correct the misusing of quantifiers and prepositions, such as changing “compare to” to “compare with”, delete the “but” in “although…but…”, changing the “test with” to “test of”, changing the “are conducted in the testbed” to “are conducted on the test bed”, et al.

(4)Correct the sentences written in first person plural.

(5)Correct the misusing of singular and plural numbers.

(6)Correct some inappropriate expressions, such as: “which is the case in practical industrial applications ”à” which is ubiquitous in practical industrial applications”, “has always been a research hotspot”à” captures increasing attention” et al.

(7)Correct the misusing of words: such as “reconstruct output”à”reconstructed output”, “adaptively feature learning ability”à“adaptive feature learning ability”, et al.

(8)Correct all spelling errors in the paper: such as “full-connected”à”fully connected”, “easy to overfitting” à“easy to over fit”, et al.

Point 2: The abstract does not provide the reader with information about the results obtained.

 Response 2: we have added the information about the results obtained to the abstract as follows: "The experimental result shows that the proposed DCAE model is effective in denoising and almost causes no loss of input information, while the using of global average pooling and input-corrupt training improves the anti-noise ability of the CNN model. As a result, the method combined the DCAE model and the CNN model can realize high-accuracy diagnosis even under noisy environment."

Point 3: A critical analysis of the results and explanation of trends is needed. It needs to be improved, as it is very cryptic in the current form, without numerical values for the results.

 Response 3: Thank you for pointing it out, according to the comment, the critical analyses of the results and the explanations of trends are added to each experiments in this paper:

 4.1.1. Validation of the denoising and diagnostic effects of the proposed method

“The reasons are analyzed as follows. Firstly, DCAE-1D is superior to DAE in reconstruction ability owning to the deeper structure. Secondly, DCAE-1D has much less parameters (2×(15×16+10×32×16+5×32×64) =31200) than that of DAE (2000×1000×2 =4000000), which makes it not easy to over fit under noisy conditions. In addition, under the SNR greater than 7 dB, the reconstruct errors of DCAE-1D are close to 0.001 while that of DAE are greater than 0.012, which indicates that the former almost causes no loss of input information in comparison with the latter.”

 “The diagnosis accuracy of the “AICNN-1D + DAE” combination and AICNN-1D without denoising drops to 94.23% and 91.64% respectively when SNR =2 dB. The result shows that the denoising step is significant to the accuracy improvement under low SNR conditions and DCAE is much more effective in noise reduction than DAE. In addition, other two points of the results are worth to be noticed: Firstly, under the conditions of SNR >2 dB, the diagnosis accuracy of the “AICNN-1D + DAE” combination is even lower than AICNN-1D without denoising. The reason for this phenomenon is that DAE causes information loss, which outweighs the gain of the denoising. Secondly, when SNR =2 dB, even though the diagnosis accuracy of AICNN-1D without denoising is less than the other two method combinations, but considering that there is no denoising, the accuracy of 91.64% is pretty well. It indicates that AICNN-1D has the capability of anti-noise diagnosis to some extent.”

 4.1.2. Validation of input-corrupt training and global average pooling

“The results prove that the input-corrupt training and the global average pooling are both effective in improving the anti-noise ability of the model. On one hand, the input-corrupt training is a regularization method, on the other hand, the application of global average pooling dramatically reduces the number of model parameters. Both of them prevent the model from overfitting, which improves the performance in the diagnosis of noisy signals. However, the accuracies of the three models are very close under high SNR conditions (SNR > 9 dB). For example, the accuracies of the three models are 99.78%, 99.75% and 99.78% when SNR= 12 dB. The reason is that all the three models are not easy to over fit in processing the signals with little noise.”

4.1.3 Optimization of the DACE-1D and AICNN-1D

Add some numerical values of the results.

4.14. Comparisons with the existing models

Add another model named WDCNN to compare with the proposed method, and rewrite the expression of the results with more numerical values and analyzation:

“Without denoising, both of the SDAE-based model and WDCNN are inferior to the proposed method under the conditions of low SNR. For instance, when SNR< 3 dB, the accuracies of the two models are less than 95%, but that of the proposed method is greater than 96.65%, especially, when SNR= -2 dB, the accuracies of the two models are 72.14% and 82.13% respectively. With the denoising by DCAE-1D, the accuracies of the SDAE-based model and WDCNN are both obviously improved, especially under the conditions of low SNR. As an example, the accuracy of the SDAE-based model rises from 72.34% to 89.12%, and that of WDCNN rises from 82.13% to 92.24% when SNR= -2dB. The results imply that in order to diagnosis accurately under low SNR conditions, the denoising is indispensable. In addition, with the same denoising preprocessing, the proposed AICNN-1D still performs better than the SDAE-based model and WDCNN obviously when SNR< 3 dB. It proves that AICNN-1D is superior to the other two models in anti-noise ability, for the parameters of AICNN-1D are much less than the other two models and the using of input-corrupt training.”

4.2.3 Feature learning visualization

An analysis of the result is added: “It is noteworthy that, the learned features of a few TCF and TWF samples in the last convolutional layer overlap with the set of Normal features, which indicates that the TCF and TWF samples may be misclassified into the label of Normal under the low SNR conditions.”

Point 4: The authors should change all the sentences in the text which are written in the first plural (line 72, 76, 79, 84, 86, 93, 96, 99, 107, 176, 225, 260, 307, 321, 339, 344, 417, 432 and 434. 

Response 4: All the sentences written in first person plural have been corrected.

Point 5: In lines 244-245, “The vibration signals are acquired by acceleration sensor installed on bearing seat in vertical direction…..” The authors don’t give any information about the accelerometers. Are they triaxial or monoaxial? How the specific position was selected? Why vertical? What is the specific position? Give a photo.

Response 5:

5-1 Information about accelerometers:  the information of the sensors is added in “4.1. Case1: bearing diagnosis experiment”: “A monoaxial piezoelectric acceleration sensor is used to acquire vibration signals, and the sensitivity of the sensor is 99.8 mv/g”, and in “Case2: gearbox diagnosis experiment”: “The acceleration sensor and its sampling frequency are the same as the bearing experiment configuration”

5-2 The selection of specific position and direction: The reasons of the selection of specific position and direction are added in “4.1. Case1: bearing diagnosis experiment”: Considering that the sensor should be close to the vibration source and in the direction of maximum vibration, it is installed on the bearing seat in the vertical direction as shown in Figure 5”, and in “Case2: gearbox diagnosis experiment”: “Considering the up cover of the gearbox is smooth in surface and receives the vibration of the gear directly, the sensor is installed on the center of the up cover of gearbox in the vertical direction, as shown in Figure 5.”

The Figure 5 is replaced with a new photo that contains the installation information of the sensors.

Point 6: In lines 248-249 the authors mention “As shown in Figure 6, each fault has three different fault width of 0.18mm, 0.36mm and 0.54mm respectively….”. Suggested to the authors to give some information about the faults. How that damages were caused. Maybe a photo. The same in Case study 2.

Response 6:  The detailed information of the faults has been added to the paper.

In Case study 1 about the bearing faults (4.1. Case1: bearing diagnosis experiment): “Wire cutting is used to process the normal bearing to simulate different faults. As shown in Figure 6, each fault has three different fault widths of 0.18mm, 0.36mm and 0.54mm and has the same fault depth of 0.3 mm.”

In Case study 2 about the gear faults (4.2. Case2: gearbox diagnosis experiment): “Five normal gears are used in the experiment named as gear A~E respectively. Electrical discharge machining is employed to process a single tooth on gear B to simulate tooth pitting fault, while wire cutting is utilized to process a single tooth on gear C and D to simulate tooth crack and tooth break fault. Besides, all the teeth on gear E are ground by gear grinding machine to simulate tooth wear fault.”;

The suggestion that giving a photo of the fault-making processing is really useful. But I am sorry for not giving the photos, because the fault bearings and gears are made by the professional manufacturer according our requirements and we forgot to ask them to take the photos. I will remember it and take the photos next time, and thank you for the suggestion.

Point 7: In line lines 243-244 the authors mention that “The radial load is set to 0 N and axial load is set to 0 N·m, while the motor speed is set to 1500 rpm”. All the measurements in the experimental system were carried out with any load? Why?

Response 7: In this paper, the method is indeed validated by experiments carried out without any load. The reason is as follows: Firstly, we want validate our method under the condition of constant radial and axial load, and the no-load condition is one of the typical cases. Secondly, because of the design defect of the test-bed, the radial force load and the axial torque load can only be keep constant but cannot be measured, so honestly, it is also a compromise.

In fact, the measurements under specific and constant radial and axial load are also taken on the testbed, and the datasets are used to test the proposed method. The experimental results are even better than that of no-load condition, the spectrum analysis indicates that under the condition of radial and axial load, the fault characteristic frequencies are more prominent compared to the no-load condition. So no-load condition is a harder case for diagnosis than other constant-load conditions, this is another reason why we choose the no-load settings.

In addition, a comment of the issue is added in the conclusion: “In this paper, the load and rotating speed are kept constant during experiment, which is a simplification of the actual conditions in industrial applications. It will be more valuable to extend the proposed method to the cases with more complex conditions (e.g. variable speed and load) in the future.”

Point 8: Conclusion: The Conclusions are very poor and are not suitable, should give more useful conclusions. Should include more numerical values for the results. The authors should rewrite more analytically this section.

 Response 8: The Conclusion section has been rewritten more analytically and with more numerical values for the results. The rewritten conclusions are as follows:

 “This paper proposes a combined method of 1-D denoising convolutional autoencoder named DCAE-1D and 1-D convolutional neural network named AICNN-1D, which aims at addressing the fault diagnosis problem under noisy environment. Compared with the existing method that only improves the anti-noise ability [21,22,30], the extra noise reduction by DACE-1D is introduced in the combined method. With the denoising of raw signals, AICNN-1D with anti-noise improvements is then used for diagnosis. The method combination is validated by the bearing and gearbox datasets mixed with Gaussian noise. Through the analysis of the experimental results in section4, conclusions are drawn as follows:

The proposed DCAE-1D with deep structure of three convolutional and three transposed convolutional layers is much more effective in denoising than the traditional DAE. The reconstruct errors of DCAE-1D in bearing and gearbox datasets are only 0.24 and 0.0109 even when the SNR= -2 dB, while that of DAE are 0.0559 and 0.01824 respectively. Besides, under the high SNR (> 4dB) conditions, the reconstruct errors are close to 0, which indicates that DACE-1D almost causes no loss of input information.

With the denoising of DCAE-1D, the diagnosis accuracies of AICNN-1D are 96.65% and 97.25% respectively in bearing and gearbox experiments even when SNR = -2 dB. When SNR> -1 dB, the diagnosis accuracies are greater than 99% in the two experiments. The result indicates that the proposed method combination realizes high diagnosis accuracy under low SNR conditions. In the bearing experiment, the AICNN-1D with “clean-input training + global average pooling” and “input-corrupt + fully-connected layers” drops to 93.54% and 94.26% in accuracy when SNR = -2 dB, compared with the proposed AICNN-1D. The results demonstrate that the using of input-corrupt training and global average pooling is effective in improving the anti-noise ability. Compared with the existing methods of the SDAE-based model and WDCNN presented in 2017, the proposed method shows its superiority under low SNR conditions. For instance, the diagnosis accuracies of the SDAE-based model and WDCNN are 72.34% and 82.13% when SNR= -2 dB in the bearing experiment. With the denoising by DCAE-1D, the diagnosis accuracies of the two models rise to 89.12% and 92.24% respectively, proving that both the denoisng of input signals and the anti-noise ability of the model are indispensable in accurate diagnosis under low SNR conditions.

In this paper, the load and rotating speed are kept constant during experiment, which is a simplification of the actual conditions in industrial applications. It will be more valuable to extend the proposed method to the cases with more complex conditions (e.g. variable speed and load) in the future. Besides, this paper focus on the diagnosis problem of industrial applications and validate the method with massive experimental data. However, the faulty sets in industrial systems are generally not readily to acquire. As the possible solutions, two kinds of methods are worth noticing: data augmentation and data fusion. The former focus on generating more samples on the basis of existing samples, such as data overlapping [21] and GAN (Generative Adversarial Network) methods [33]. The latter studies fusing the data of multiple kinds of sensors, which can be grouped into three main approaches: data-level fusion [34], feature-level fusion[35] and decision-level fusion [36].”

Reviewer 2 Report

The paper presents the fault diagnosis method of rotating machinery based on deep learning CNN method. The application of CNN for 1D data is still promising.
The paper is well presented, sufficient contents for journal publication. However, there are some minor comments:
1. What the meaning of '/' symbol in Table 2?
2. It is mentioned in Lines 172-174 that the AICNN-1D is used after de-noised signals output by DCAE-1D. However, this is not reflected in Figure 4 where there is no relation (linked arrow) between DCAE-D block and AICNN-1D block. Could you please clarify this?
3. Could you please indicate which are the existing models and which is the proposed model in Figure 14?
4. Could you please also set the grid off for Figures 8-11, 13, and 14.
5. I don't think that the Abbreviations in line 436 is needed, because you have defined all the abbreviation in the paper.
6. Please also provide the comparison of computational time between the existing models/methods and proposed model/method.

Author Response

Dear Reviewer:

Thank you for your comments concerning our manuscript entitled “Fault diagnosis of rotating machinery under noisy environment based on 1-D convolutional autoencoder and 1-D convolutional neural network” (ID: sensors-430233). Those comments are all valuable and very helpful for revising and improving our paper, as well as the important guiding significance to our researches. We have studied comments carefully and have made correction which we hope meet with approval. Revised portion are marked in red in the paper. The main corrections in the paper and the responds to your comments are as flowing:

Point 1: What the meaning of '/' symbol in Table 2?

Response 1: The symbol ‘/’ originally means that the layer has no such parameter or term. Now we have replaced the symbol ‘/’ with the word “None”, which is clearer.

Point 2: It is mentioned in Lines 172-174 that the AICNN-1D is used after de-noised signals output by DCAE-1D. However, this is not reflected in Figure 4 where there is no relation (linked arrow) between DCAE-D block and AICNN-1D block. Could you please clarify this?

Response 2: Thank you for pointing this out and the relation of DCAE-1D and AICNN-1D is indeed not clear in the original Figure 4. For clarify this, the Figure are modified as follows:

In the “Trained Model Test” block, the “De-noised signals” is deleted and it does not affect the meaning to be expressed. As a result, the “Trained DCAE-1D” and “Trained AICNN-1D” is linked directly by an arrow, which indicates that in the test of the method, the AICNN-1D is used after the denoising by DCAE-1D. In addition, the training of DACE-1D and AICNN-1D is independent, so there is no linked arrow between the block “DCAE-1D Unsupervised Learning” and the block “AICNN-1D Supervised Learning”. 

Point 3: Could you please indicate which are the existing models and which is the proposed model in Figure 14?

Response 3: We have added another model named WDCNN to compare with the proposed model. And have modified the original figure to indicate the existing models and the proposed model, the improved figure is shown as follows:

Point 4: Could you please also set the grid off for Figures 8-11, 13, and 14.

Response 4: The grid of Figures 8-11, 13 and 14 has been set off. 

Response 6: The comparison of computational time between the existing methods and the proposed method has been added in “4.14. Comparisons with the existing models”:

“The comparison results of computational time between the existing models and the proposed models are shown in Table 10. It can be seen that the training of DCAE-1D and AICNN-1D costs 106.88 ± 3.75 s and 53.25 ± 2.79 s respectively, and the testing costs 0.79 ± 0.05 s and 0.42 ± 0.06 s. Compared with the existing deep method of the SDAE-based model and WDCNN, which costs 98.64 ± 2.58 s and 0.71 ± 0.12 s in training respectively and costs 0.71 ± 0.12 s and 0.84 ± 0.08 s in testing, the proposed method combination consumes slightly more time in training and testing. In addition, the shallow models of BP neural network and SVM are much less time-consuming in both training and testing.”

Table. Computational time of the proposed models and the existing models.

Method

Training time (5000×100 samples)

Testing time(10000 samples)

DCAE-1D + AICNN-1D

(106.88 ±   3.75) s + (53.25 ± 2.79) s

(0.79 ±   0.05) s + (0.42 ± 0.06) s

SDAE

98.64 ± 2.58   s

0.71 ± 0.12   s

WDCNN

126.27 ±   5.24 s

0.84 ± 0.08   s

BP

40.43 ± 0.93   s

0.36 ± 0.06   s

SVM

63.56 ± 0.78   s

0.57 ± 0.11   s

Reviewer 3 Report

The paper is a pleasure to read and it is welly described and presented.

  the paper fails in comparing the obtained results and the proposed approach with related works.  It could have been better if the proposed approach was compared with more other recent approaches. In 4.14. Comparisons with the existing models section, it is clearly evident that, BP neural network and support vector machine are also used for comparison beside SDAE approach but no mention about where they were used before and absence of references too.

Even though an introduction provide background and relevant references, but still it is not sufficient. For example, in image recognition field, there is already DCAE. you can explain what your novelty is.

For making noisy samples, the authors used Gaussian noise and SNR variation. Does SNR variation is effected in whole area? Explain how to make it in more detailed. If SNR varies in whole area, I suggest that the authors make additional experiments with partially varying SNR and randomly effected dataset.

minor suggestions

- English language and style are not fine and minor spell check is required.

- In case of research design and description of methods, it was eloquently written.

- Results are clearly presented.

- As mention in first point, there are so many grammatical errors in the sentence and words. So even the conclusion part tries to support the results, it still lacks of uniformity and flow of information in the sentences.

- A lot of comma (,) uses when (.) is needed. Some sentences should be divided to make context clear.

- Errors like (full-connected -> fully-connected)

- Please, check your punctuation carefully.

- Avoid mentioning references like Paper[26].

Author Response

Dear Reviewer:

Thank you for your comments concerning our manuscript entitled “Fault diagnosis of rotating machinery under noisy environment based on 1-D convolutional autoencoder and 1-D convolutional neural network” (ID: sensors-430233). Those comments are all valuable and very helpful for revising and improving our paper, as well as the important guiding significance to our researches. We have studied comments carefully and have made correction which we hope meet with approval. Revised portion are marked in red in the paper. The main corrections in the paper and the responds to your comments are as flowing:

Point 1: the paper fails in comparing the obtained results and the proposed approach with related works.  It could have been better if the proposed approach was compared with more other recent approaches. In 4.14. Comparisons with the existing models section, it is clearly evident that, BP neural network and support vector machine are also used for comparison beside SDAE approach but no mention about where they were used before and absence of references too.

Response 1:  According to the comment, we have rewritten the “4.14. Comparisons with the existing models” and the improvements are as follows:

1-1 we add a model named WDCNN that aiming at the anti-diagnosis problem to compare with the proposed method, which is proposed in 2017.

“The second one is named WDCNN proposed by Zhang et al. in 2017 [22]. WDCNN adopts five 1-D convolutional layers, five max-pooling layers and two fully-connected layers. In order to improve the anti-noise ability, AdaBN layers and wide first-layer kernels are employed. The experimental results are shown in Figure 12.”

1-2 we supplement the references of BP neural network and support vector machine in the paper.

“The classic shallow models of the BP neural network [31] and the SVM [32] are also adopted to compare with the proposed method. The input, hidden and output dimensions of the BP neural network is 2000, 1000 and 10, and the sigmoid function is used as activation function. The SVM model uses Gaussian kernel function. Various configurations for the BP neural network and the SVM are explored the configurations that achieved the best performance are empirically selected.”

1-3 The expression of the comparison results has been rewritten with more numerical values and analyses of the results.

“Without denoising, both of the SDAE-based model and WDCNN are inferior to the proposed method under the conditions of low SNR. For instance, when SNR< 3 dB, the accuracies of the two models are less than 95%, but that of the proposed method is greater than 96.65%. With the denoising by DCAE-1D, the accuracies of the SDAE-based model and WDCNN are both obviously improved, especially under the conditions of low SNR. As an example, the accuracy of the SDAE-based model rises from 72.34% to 89.12%, and that of WDCNN rises from 82.13% to 92.24% when SNR= -2dB. The results imply that in order to diagnosis accurately under low SNR conditions, the denoising is indispensable. In addition, with the same denoising preprocessing, the proposed AICNN-1D still performs better than the SDAE-based model and WDCNN obviously when SNR< 3 dB. It proves that AICNN-1D is superior to the other two models in anti-noise ability, for the parameters of AICNN-1D are much less than the other two models and the using of input-corrupt training.”

“Various configurations for the BP neural network and the SVM are explored the configurations that achieved the best performance are empirically selected. As shown in Figure 12, even with the denoising by DCAE-1D, these two models both perform much worse than the three models above under all SNR conditions. For example, the accuracies of the BP neural network and the SVM are only 87.68% and 92% respectively even when SNR= 12 dB. The results indicate that the shallow models cannot extract discriminative features from vibration signals and cannot be regarded as effective end-to-end methods.

1-4: we also add the comparison of the computational time between the proposed method and the existing methods.

“The comparison results of computational time between the existing models and the proposed models are shown in Table 10. It can be seen that the training of DCAE-1D and AICNN-1D costs 106.88 ± 3.75 s and 53.25 ± 2.79 s respectively, and the testing costs 0.79 ± 0.05 s and 0.42 ± 0.06 s. Compared with the existing deep method of the SDAE-based model and WDCNN, which costs 98.64 ± 2.58 s and 0.71 ± 0.12 s in training respectively and costs 0.71 ± 0.12 s and 0.84 ± 0.08 s in testing, the proposed method combination consumes slightly more time in training and testing. In addition, the shallow models of BP neural network and SVM are much less time-consuming in both training and testing.”

Table 10. Computational time of the proposed models and the existing models.

Method

Training time (5000×100 samples)

Testing time(10000 samples)

DCAE-1D + AICNN-1D

(106.88 ±   3.75) s + (53.25 ± 2.79) s

(0.79 ±   0.05) s + (0.42 ± 0.06) s

SDAE

98.64 ± 2.58   s

0.71 ± 0.12   s

WDCNN

126.27 ±   5.24 s

0.84 ± 0.08   s

BP

40.43 ± 0.93   s

0.36 ± 0.06   s

SVM

63.56 ± 0.78   s

0.57 ± 0.11   s

Point 2: Even though an introduction provides background and relevant references, but still it is not sufficient. For example, in image recognition field, there is already DCAE. You can explain what your novelty is.

Response 2: Thank you for pointing it out, we have added some researches in the area of image denoising in “1. Introduction”:

“In the area of image denoising, the deep learning methods are widespread. In 2018 [23], a method based on deep CNN is proposed, which is trained by the image with non-fixed noise masks. Mao et al. proposed a deep DCAE based method aiming at image denoising and super-resolution [24].”

And the novelty of the proposed method is explained subsequently:

“However, these methods are only applied in denosing of 2-D or 3-D image, and to the best of our knowledge, no research of similar methods aiming at the denoising of 1-D vibration signal has been reported so far.”

Point 3: For making noisy samples, the authors used Gaussian noise and SNR variation. Does SNR variation is effected in whole area? Explain how to make it in more detailed.

Response 3:  The SNR variation is effected in whole area. A detailed explanation of making the noisy datasets is added in “3.1. DCAE-1D”:

“To compose the noisy signals with different SNR, the noise with different power are superposed to the same original signal respectively in whole area. The noise power is calculated according to the power of the original signal and the specified SNR value. Then each composited noisy signal is divided into multi-segments, which compose an original dataset with the specified SNR. Each original dataset is divided into two parts in a certain proportion, namely the training set and the testing set. Then all the training sets are gathered and shuffled to form the training dataset for training DACE-1D, while all the testing sets are respectively used for model testing.”

Point 4: If SNR varies in whole area, I suggest that the authors make additional experiments with partially varying SNR and randomly effected dataset.

Response 4: This is really an insightful suggestion and we have made another experiment with partially varying SNR in “4.2.2. Validation of partial varying SNR situation”:

“In the previous experiments, the denoising of DCAE-1D and the diagnosis of AICNN-1D are tested by several datasets with different SNR. Although the SNR varies between different datasets, it still keeps constant in the same dataset. Furthermore, in order to simulate the real noise that continuous changing in industrial environment as exactly as possible, the dataset with partially varying SNR is used to test the denoising of DCAE-1D and the diagnosis of AICNN-1D. To be specific, the dataset is obtained through the following steps: Firstly, divide each sample of the noise-free test set into 10 segments in sequence, and then assign each segment a random SNR value between -2 dB to 12 dB. Secondly, the Gaussian noise of specific power is superposed to each segment according to the power and the assigned SNR value of the signal segment. Finally, recombine every 10 noisy segments into one sample in the original order.

The dataset is generated 20 times through the steps above. Correspondingly, each generated dataset is used to test the denoising of DCAE-1D and the diagnosis of AICNN-1D independently. The statistical results of the 20 test are as follows: the reconstruct error of DCAE-1D is 0.0063 ± 0.0005 and the accuracy of AICNN-1D is 99.34% ± 0.17%. The results demonstrate that the method combination of DACE-1D and AICNN-1D is also effective under the situation of partial varying SNR.”

Point 5: English language and style are not fine and minor spell check is required.

Response 5: Thank you for pointing it out. The language has been thoroughly checked and the improvements are listed as follows:

(1)Add definite articles in places, or change some indefinite articles to definite articles.

(2)Correct the misusing of punctuation, such as changing the comma (,) to (.), or changing the comma (;) to (.) or (,), et al.

(3)Correct the misusing of quantifiers and prepositions, such as changing “compare to” to “compare with”, delete the “but” in “although…but…”, changing the “test with” to “test of”, changing the “are conducted in the testbed” to “are conducted on the test bed”, et al.

(4)Correct the sentences written in first person plural.

(5)Correct the misusing of singular and plural numbers.

(6)Correct some inappropriate expressions, such as: “which is the case in practical industrial applications ”to” which is ubiquitous in practical industrial applications”, “has always been a research hotspot” to ” captures increasing attention” et al.

(7)Correct the misusing of words: such as “reconstruct output” to ”reconstructed output”, “adaptively feature learning ability” to “adaptive feature learning ability”, et al.

(8)Correct all spelling errors in the paper: such as “full-connected” to ”fully connected”, “easy to overfitting”  to “easy to over fit”, et al.

Point 6: As mention in first point, there are so many grammatical errors in the sentence and words. So even the conclusion part tries to support the results, it still lacks of uniformity and flow of information in the sentences.

Response 6: The Conclusion section has been rewritten more analytically and with more numerical values for the results. The rewritten conclusions are as follows:

“This paper proposes a combined method of 1-D denoising convolutional autoencoder named DCAE-1D and 1-D convolutional neural network named AICNN-1D, which aims at addressing the fault diagnosis problem under noisy environment. Compared with the existing method that only improves the anti-noise ability [21,22,30], the extra noise reduction by DACE-1D is introduced in the combined method. With the denoising of raw signals, AICNN-1D with anti-noise improvements is then used for diagnosis. The method combination is validated by the bearing and gearbox datasets mixed with Gaussian noise. Through the analysis of the experimental results in section4, conclusions are drawn as follows:

The proposed DCAE-1D with deep structure of three convolutional and three transposed convolutional layers is much more effective in denoising than the traditional DAE. The reconstruct errors of DCAE-1D in bearing and gearbox datasets are only 0.24 and 0.0109 even when the SNR= -2 dB, while that of DAE are 0.0559 and 0.01824 respectively. Besides, under the high SNR (> 4dB) conditions, the reconstruct errors are close to 0, which indicates that DACE-1D almost causes no loss of input information.

With the denoising of DCAE-1D, the diagnosis accuracies of AICNN-1D are 96.65% and 97.25% respectively in bearing and gearbox experiments even when SNR = -2 dB. When SNR> -1 dB, the diagnosis accuracies are greater than 99% in the two experiments. The result indicates that the proposed method combination realizes high diagnosis accuracy under low SNR conditions. In the bearing experiment, the AICNN-1D with “clean-input training + global average pooling” and “input-corrupt + fully-connected layers” drops to 93.54% and 94.26% in accuracy when SNR = -2 dB, compared with the proposed AICNN-1D. The results demonstrate that the using of input-corrupt training and global average pooling is effective in improving the anti-noise ability. Compared with the existing methods of the SDAE-based model and WDCNN presented in 2017, the proposed method shows its superiority under low SNR conditions. For instance, the diagnosis accuracies of the SDAE-based model and WDCNN are 72.34% and 82.13% when SNR= -2 dB in the bearing experiment. With the denoising by DCAE-1D, the diagnosis accuracies of the two models rise to 89.12% and 92.24% respectively, proving that both the denoisng of input signals and the anti-noise ability of the model are indispensable in accurate diagnosis under low SNR conditions.

In this paper, the load and rotating speed are kept constant during experiment, which is a simplification of the actual conditions in industrial applications. It will be more valuable to extend the proposed method to the cases with more complex conditions (e.g. variable speed and load) in the future. Besides, this paper focus on the diagnosis problem of industrial applications and validate the method with massive experimental data. However, the faulty sets in industrial systems are generally not readily to acquire. As the possible solutions, two kinds of methods are worth noticing: data augmentation and data fusion. The former focus on generating more samples on the basis of existing samples, such as data overlapping [21] and GAN (Generative Adversarial Network) methods [33]. The latter studies fusing the data of multiple kinds of sensors, which can be grouped into three main approaches: data-level fusion [34], feature-level fusion [35] and decision-level fusion [36].”

Point 7: A lot of comma (,) uses when (.) is needed. Some sentences should be divided to make context clear.

Response 7: We have checked the paper throughout and divided some long sentences to short sentences, making the context clearer.

A lot of comma (,) and (;) have been replaced with (.).

Point 8: Errors like (full-connected -> fully-connected)

Response 8: we have corrected all the spelling errors in the paper: such as “full-connected” to ”fully connected”, “easy to overfitting” to “easy to over fit”, et al.

Point 9: Please, check your punctuation carefully.

Response 9: We have checked the punctuation throughout the paper, and the misusing of comma (,), (;) and (:) have been corrected in places.

Point 10: Avoid mentioning references like Paper [26].

Response 10: We have checked all references through the paper, and rewrite the reference as “Vincent et al. point out that the random-corruption can create noise for the input signals, which makes the trained model performs better even when the training set and testing set have different distributions [26].”

Reviewer 4 Report

This paper has presented a fault diagnosis method based on 1D-DCAE and 1D-CNN, specifically dealing with noisy signals. It is in general easy to follow, although there are some grammar errors in places. Please also check on the format of the symbols and equations, which seems strange.

One technical question is: The validation is based on mixing the signals with Gaussian noise, making different SNRs and keeping the SNR constant in the same test. Since the authors have the testbed available to collect signals, wouldn't it be more credible if the method can be tested based on data with 'real' background noise, e.g. running another motor, rather than synthetic noise adding?

If the noise in signals part was aimed for industrial applications, industrial systems also do not normally have readily faulty sets for classification model training. Rather, more CBM methods are relying on Novelty detection. Please comment on this point also.

Furthermore, for the results section, to be more rigorous, please show at least one example of the reconstruction results after DAE and DCAE-1D along with Figure 7, and at least one example of the classification result of AICNN-1D, e.g. the confusion matrix, before or after Figure 9.

All the figures showing accuracies could be better presented also.

Author Response

Dear Reviewer:

Thank you for your comments concerning our manuscript entitled “Fault diagnosis of rotating machinery under noisy environment based on 1-D convolutional autoencoder and 1-D convolutional neural network” (ID: sensors-430233). Those comments are all valuable and very helpful for revising and improving our paper, as well as the important guiding significance to our researches. We have studied comments carefully and have made correction which we hope meet with approval. Revised portion are marked in red in the paper. The main corrections in the paper and the responds to your comments are as flowing:

Point 1: This paper has presented a fault diagnosis method based on 1D-DCAE and 1D-CNN, specifically dealing with noisy signals. It is in general easy to follow, although there are some grammar errors in places. Please also check on the format of the symbols and equations, which seems strange.

Response 1: Thank you for pointing out the problems of grammar errors, the symbols and the equations. We have carefully checked the paper thoroughly and the improvements are made as follows:

1-1: With respect to the grammar errors:

(1)Add definite articles in places, or change some indefinite articles to definite articles.

(2)Correct the misusing of punctuation, such as changing the comma (,) to (.), or changing the comma (;) to (.) or (,), et al.

(3)Correct the misusing of quantifiers and prepositions, such as changing “compare to” to “compare with”, delete the “but” in “although…but…”, changing the “test with” to “test of”, changing the “are conducted in the testbed” to “are conducted on the test bed”, et al.

(4)Correct the sentences written in first person plural.

(5)Correct the misusing of singular and plural numbers.

(6)Correct some inappropriate expressions, such as: “which is the case in practical industrial applications ” to ” which is ubiquitous in practical industrial applications”, “has always been a research hotspot” to ” captures increasing attention” et al.

(6)Correct the misusing of words: such as “reconstruct output” to ”reconstructed output”, “adaptively feature learning ability” to “adaptive feature learning ability”, et al.

(7)Correct all spelling errors in the paper: such as “full-connected” to ”fully connected”, “easy to overfitting” to “easy to over fit”, et al.

1-2: With respect to the format of the symbols and equations, we have made improvements according the classic paper and the requirements of the Journal:

A: The symbols has been corrected according to the classic paper.

B:The format problems have been corrected. Such as correcting the italic brackets “()” to “()”, italic functions such as “max/avg” to “max/avg”.

Point 2: One technical question is: The validation is based on mixing the signals with Gaussian noise, making different SNRs and keeping the SNR constant in the same test. Since the authors have the testbed available to collect signals, wouldn't it be more credible if the method can be tested based on data with 'real' background noise, e.g. running another motor, rather than synthetic noise adding?

Response 2: This is a really insightful question and what we wanted to do. However, when we did the experiments, two problems were arisen: Firstly, when we designed the testbed originally, we had taken the influence of the operation environment into consideration. So the testbed is designed with a very heavy base, weighing 800kg. As a result, when we knock on the testbed with a hammer or run another motor on the test bed, the vibration signals of the gear and the bearing are almost unaffected. Secondly, we find that the slight noises added to the vibration signals of the bearing and the gear are hard to quantify, in addition, we find that we can hardly know the distribution of the slight noise.

Besides, because of your comments, we have tried to find the answer why most of researchers using Gaussian noise to simulate the noise in industrial environments. On reason is that: in real environments, noise is often not caused by a single source, but complex of many different sources. Considering the real noise as a combination of many random variables with different probability distributions, and each random variable is independent, then according to Central Limit Theorem, their normalized sum tends to be a Gauss distribution as the number of noise sources increases.

Finally, we add another experiment in “4.2.2. Validation of partial varying SNR situation”, to simulate the real noise that continuous changing in industrial environment as exactly as possible:

“In the previous experiments, the denoising of DCAE-1D and the diagnosis of AICNN-1D are tested by several datasets with different SNR. Although the SNR varies between different datasets, it still keeps constant in the same dataset. Furthermore, in order to simulate the real noise that continuous changing in industrial environment as exactly as possible, the dataset with partially varying SNR is used to test the denoising of DCAE-1D and the diagnosis of AICNN-1D. To be specific, the dataset is obtained through the following steps: Firstly, divide each sample of the noise-free test set into 10 segments in sequence, and then assign each segment a random SNR value between -2 dB to 12 dB. Secondly, the Gaussian noise of specific power is superposed to each segment according to the power and the assigned SNR value of the signal segment. Finally, recombine every 10 noisy segments into one sample in the original order.

The dataset is generated 20 times through the steps above. Correspondingly, each generated dataset is used to test the denoising of DCAE-1D and the diagnosis of AICNN-1D independently. The statistical results of the 20 test are as follows: the reconstruct error of DCAE-1D is 0.0063 ± 0.0005 and the accuracy of AICNN-1D is 99.34% ± 0.17%. The results demonstrate that the method combination of DACE-1D and AICNN-1D is also effective under the situation of partial varying SNR.”

Point 3: If the noise in signals part was aimed for industrial applications, industrial systems also do not normally have readily faulty sets for classification model training. Rather, more CBM methods are relying on Novelty detection. Please comment on this point also.

Response 3: A comment of this point has added to the “5. Conclusions”: “Besides, this paper focus on the diagnosis problem of industrial applications and validate the method with massive experimental data. However, the faulty sets in industrial systems are generally not readily to acquire. As the possible solutions, two kinds of methods are worth noticing: data augmentation and data fusion. The former focus on generating more samples on the basis of existing samples, such as data overlapping [21] and GAN (Generative Adversarial Network) methods [33]. The latter studies fusing the data of multiple kinds of sensors, which can be grouped into three main approaches: data-level fusion [34], feature-level fusion [35] and decision-level fusion [36].”

Point 4: Furthermore, for the results section, to be more rigorous, please show at least one example of the reconstruction results after DAE and DCAE-1D along with Figure 7, and at least one example of the classification result of AICNN-1D, e.g. the confusion matrix, before or after Figure 9.

4-1 An example of the reconstruction results after DAE and DCAE-1D are added in Figure 8, and the results are analysed as follows:

Figure 8 shows the original signal, the noisy signal, the de-noised signal of DCAE-1D and the de-noised signal of DAE under the condition of SNR = -1 dB and roller fault (0.54mm). It can be seen that the fault characteristics in original signal are masked by the added Gaussian noise. After the denoising by DCAE-1D or DAE, the noise mixed in the original signal is obviously reduced. However, the noise cannot be eliminated completely, and the result shows that the de-noised signal of DCAE-1D leaves much less noise than that of DAE.”

4-2 An example of the classification result of AICNN-1D is added as confusion matrix in Figure 10 and the result is analysed as follows:

Figure 10 presents the confusion matrix of the diagnosis result under the condition of SNR = -2 dB and the overall accuracy is 96.65%. It shows that the accuracy of Normal condition is the lowest one, which is 94.5%. Besides, the IF-0.018, OF-0.018 and RF-0.018, regarded as slight fault conditions, are 94.8%, 95.4% and 95.7% in accuracy respectively, which are lower than the rest health conditions. The results demonstrate that the Normal and slight fault conditions are more likely to be misclassified under low SNR conditions. Furthermore, it can be found from Figure 10 that the Normal condition is likely to be misclassified into the slight fault conditions and vice versa. As an instance, the misclassification rates of Normal to IF-0.018, Normal to OF-0.018 and Normal to RF-0.018 are 1.7%, 1.8% and 1.3% respectively and that of Normal to other conditions are lower obviously.”

Point 5: All the figures showing accuracies could be better presented also.

Response 5: All the figures showing accuracies are improved:

(1)  In order to better distinguish different curves, the width of the curves is reduced.

(2) The grid of the figures has been set off for better presented.

Round  2

Reviewer 1 Report

The authors responded to all my questions and took into account all my suggestions.

Author Response

Dear Reviewer:

Point 1:  The authors responded to all my questions and took into account all my suggestions.

Response 1:  Thank you very much for your comments and suggestions. They are really insightful and they are very helpful in improving  the quality of our paper.

Reviewer 4 Report

The response to reviewer is satisfactory in general. There exists in the current version mainly minor formatting issues, e.g. the fonts of the symbols and equations should be unified; some figures' quality could be improved for enhancing readability, etc.

Author Response

 Dear Reviewer:

Thank you for your comments concerning our manuscript entitled “Fault diagnosis of rotating machinery under noisy environment based on 1-D convolutional autoencoder and 1-D convolutional neural network” (ID: sensors-430233). We have carefully checked the deficiencies that you point out. The main corrections and the responds to your comments are as following:

 Point 1: There exists in the current version mainly minor formatting issues, e.g. the fonts of the symbols and equations should be unified;

 Response 1: Thank you for pointing out the problem. The corrections of the symbols and equations are as follows:

(1)   The fonts of the symbols and equations have been united. (line 139, line 143, Equation 5).

(2)   The constants in the equations are changed from italic to normal. (Equation1, 2 and 5).

Point 2: some figures' quality could be improved for enhancing readability, etc.

Response 2: We have improved the resolution of the image to ensure that the resolution of each image is above 400 dpi.
